# Diffusion-Inspired Truncated Sampler for Text-Video Retrieval

**Jiamian Wang**[1†], **Pichao Wang**[2*], **Dongfang Liu**[1], **Qiang Guan**[3], **Sohail Dianat**[1],
**Majid Rabbani**[1], **Raghuveer Rao**[4], **Zhiqiang Tao**[1†]
[1]Rochester Institute of Technology, [2]Amazon,
[3]Kent State University, [4]DEVCOM Army Research Laboratory

## Abstract

Prevalent text-to-video retrieval methods represent multimodal text-video data in a joint embedding space, aiming at bridging the relevant text-video pairs and pulling away irrelevant ones. One main challenge in state-of-the-art retrieval methods lies in the modality gap, which stems from the substantial disparities between text and video and can persist in the joint space. In this work, we leverage the potential of Diffusion models to address the text-video modality gap by progressively aligning text and video embeddings in a unified space. However, we identify two key limitations of existing Diffusion models in retrieval tasks: The $\mathcal{L}_2$ loss does not fit the ranking problem inherent in text-video retrieval, and the generation quality heavily depends on the varied initial point drawn from the isotropic Gaussian, causing inaccurate retrieval. To this end, we introduce a new Diffusion-Inspired Truncated Sampler (DITS) that jointly performs progressive alignment and modality gap modeling in the joint embedding space. The key innovation of DITS is to leverage the inherent proximity of text and video embeddings, defining a truncated diffusion flow from the fixed text embedding to the video embedding, enhancing controllability compared to adopting the isotropic Gaussian. Moreover, DITS adopts the contrastive loss to jointly consider the relevant and irrelevant pairs, not only facilitating alignment but also yielding a discriminatively structured embedding. Experiments on five benchmark datasets suggest the state-of-the-art performance of DITS. We empirically find that DITS can also improve the structure of the CLIP embedding space. Code is available at https://github.com/Jiamian-Wang/DITS-text-video-retrieval

## 1 Introduction

Text-video retrieval aims to match textual descriptions with relevant video content and vice versa, utilizing both modalities to improve ranking accuracy [Zhu et al., 2023]. This task is challenging due to the multi-modality gap, which represents the inherent differences between textual and visual data representations [Yang et al., 2021, Huang et al., 2023b, Sun et al., 2024, Wang et al., 2021]. Bridging this gap effectively requires advanced methodologies to extract and align semantic information from both modalities into a joint embedding space [Cheng et al., 2023, Xu et al., 2021, Pei et al., 2023].

Current state-of-the-art approaches have explored various strategies to narrow this gap. Advanced feature extraction techniques focus on capturing fine-grained details through temporal modeling [Liu et al., 2022b] and multi-granularity matching strategies, such as frame-level, patch-level, word-level, and token-level representations [Li et al., 2023b, Guan et al., 2023]. Additionally, leveraging large-scale data augmentation has been shown to enhance the capabilities of visual-language models [Wu

---

*The work does not relate to author's position at Amazon.

†Corresponding authors: Jiamian Wang (`jw4905@rit.edu`) & Zhiqiang Tao (`zhiqiang.tao@rit.edu`)

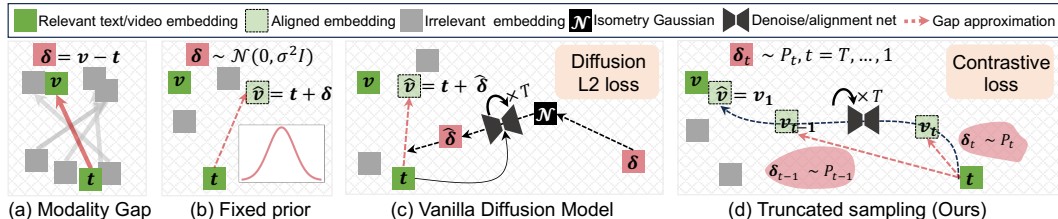

Figure 1: Multi-modality gap alignment for the text-video retrieval. **(a)** Considering the many-to-one correspondence of text-to-video, we define the modality gap stemming from the text ($\delta = \mathbf{v} - \mathbf{t}$) for uniqueness. **(b)** Fixed prior alignment posits a Gaussian distribution, which is inflexible. **(c)** Diffusion-based alignment upon $\mathcal{L}_2$ loss, which does not fit the retrieval task. The generation is heavily affected by the random samples from $\mathcal{N}(\mathbf{0}, \mathbf{I})$. **(d)** The proposed diffusion-inspired truncated sampler (DITS) aligns from $\mathbf{t}$ to $\mathbf{v}$ and gradually models the gap, guided by the contrastive loss.

et al., 2023, Wang et al., 2024b], while incorporating additional modalities like audio has also improved retrieval performance [Ibrahimi et al., 2023]. Despite the progress, the modality gap (Fig. 1 (a)) remains a persistent issue, significantly impacting the performance of retrieval systems.

A critical observation in text-video retrieval is that existing methods often struggle with the variability and complexity of the semantic alignment between text and video. This work addresses this challenge by investigating Diffusion models [Ho et al., 2019, Song et al., 2021] to bridge the modality gap. Our main contribution is developing a new Diffusion-inspired truncated sampler (DITS) for models to iteratively align video and text embeddings in a progressive manner. Unlike traditional approaches [Qiu et al., 2024, Nukrai et al., 2022] that impose fixed priors for the alignment (Fig. 1 (b)), our method starts from the text embedding and employs a truncated diffusion process to generate aligned video embeddings, addressing the inflexibility and inadequacy of fixed prior methods.

We first explore the generative capability of Diffusion models to mitigate the modality gap. Initially, we employ pre-trained text and video embeddings and model the gap using a diffusion process, where the initial state (isotropic Gaussian) represents a fixed prior, progressively denoised to align the embedding (Fig. 1 (c)). However, we found that the vanilla Diffusion model's $\mathcal{L}_2$ loss does not fit the ranking problem inherent in text-video retrieval. The $\mathcal{L}_2$ loss assumes a Gaussian distribution and minimizes the mean squared error, treating all errors equally and failing to prioritize the alignment of semantically relevant pairs over irrelevant ones. Moreover, the reverse process introduces diversity through initial Gaussian noise sampling, which impacts the generation quality and causes the mis-alignment. Therefore, it is non-trivial and necessary to tailor the Diffusion modeling from generation (where diversity is beneficial) to ranking (where precise alignment is crucial), catering to text-video retrieval. The focus of diffusion learning should be on closing relevant embeddings while pushing apart irrelevant pairs. This requires a different loss function and sampler for better alignment.

To address the above challenges, DITS leverages the inherent proximity of text and video embeddings in the joint space [Liang et al., 2022, Zhou et al., 2023] and incorporates a truncated diffusion process into the contrastive loss (depicted in Fig. 1 (d)). Different from the vanilla diffusion-based alignment, DITS adopts text embedding as a more meaningful intermediate latent state to initiate sampling, reducing the underlying variability and improving the alignment accuracy over Gaussian noises. The proposed DITS reformulates each diffusion step as modeling the modality gap and gradually controls the text-to-video alignment through the contrastive loss, acting on both aligned and video embeddings, which thus intrinsically promotes the alignment across timestamps in a more fine-grained manner. Empirical evidence shows that DITS can also enhance the structure of the CLIP embedding space. We summarize the contributions of this work as follows.

- This work studies the Diffusion model to bridge the modality gap of text-video retrieval, identifying two key limitations of the vanilla Diffusion model. A new sampler, namely DITS, is also proposed to enhance the multi-modality alignment and benefit the retrieval.

- DITS enables a new truncated diffusion process to conduct the video-text alignment progressively, starting from the intermediate latent state given by text embedding to approaching the relevant video embedding. Governed by the contrastive loss, the proposed DITS gradually models the gap distribution along each timestamp by learning from relevant/irrelevant pairs.

- Extensive experiments on five datasets (MSRVTT, LSMDC, DiDeMo, VATEX, and Charades) suggest that DITS achieves state-of-the-art performance. Empirical evidence shows that DITS also improves the structure of the CLIP embedding space.

## 2  Related Work

**Text-video Retrieval**. State-of-the-art text-video retrieval methods [Huang et al., 2023a, Gao et al., 2021, Yu et al., 2018, Deng et al., 2023, Jin et al., 2023, Croitoru et al., 2021] develop different strategies to facilitate the multi-modality alignment. Some adopt the temporal modeling [Li et al., 2023b, Liu et al., 2023, Li et al., 2023c, Han et al., 2022b] on word sequence or video frames, better extracting the semantic clues and enhancing the retrieval. Others extract multi-modality features at different granularity [Liu et al., 2022b, Chen et al., 2020b, Wang et al., 2023, Ma et al., 2022, Han et al., 2022a, Wang et al., 2021], uncovering hierarchical relationships between word sequences and video clips. Apart from the intricate model designs [Guan et al., 2023, Wu et al., 2021, Bain et al., 2021, Liu et al., 2021, Miech et al., 2021], incorporating the large-scale text-video data via augmentation emerges as a promising approach [Luo et al., 2022, Wu et al., 2023, Falcon et al., 2022], which effectively amplifies the potential of the visual-language foundation models [Chen et al., 2024, Ko et al., 2023] and benefiting the retrieval. Additionally, harnessing new modality data, such as the audio [Ibrahimi et al., 2023, Akbari et al., 2021, Lin et al., 2022, Miech et al., 2018, Liu et al., 2022a, Shvetsova et al., 2022], paves a new direction to bridge the text-video modalities, with abundant semantics clues. Recent advances [Fang et al., 2023, Wang et al., 2024a, Gao et al., 2024, Ji et al., 2023, Chun et al., 2021, Nukrai et al., 2022] study different representation forms of the multi-modality data, among which T-MASS [Wang et al., 2024a] achieves the outstanding performance. Despite the prosperity, the modality gap persists in the joint space. Distinguished from existing methods, this work studies how to learn the Diffusion model to model the modality gap in the joint embedding space, considering the substantial gap between the generation task and the ranking nature inherent to text-video retrieval.

**Diffusion models**. Diffusion models [Croitoru et al., 2023, Ho et al., 2019] define a transition from the isotropic Gaussian to the clean data with a learned diffusion process, emerging as powerful generative models for learning complex distributions. This approach has been extensively applied in image generation tasks [Yang et al., 2023], with models such as Denoising Diffusion Probabilistic Models (DDPM) [Ho et al., 2019], DDIM [Song et al., 2021], and LDM [Rombach et al., 2022] demonstrating impressive capabilities in generating high-fidelity images. Building upon these foundations, recent research [Zhang et al., 2023] has extended Diffusion models to the multimodal domain, not only enabling the generation of diverse and realistic data with distinct type (such as image, audio, etc.) [Yang et al., 2023], but also achieving the transition across varied modalities (*e.g.*, text-to-image, text-to-video, etc.) [Ghosh et al., 2024]. Nevertheless, leveraging the Diffusion model to the ranking tasks such as the text-video retrieval is non-trivial due to the significant difference in data, evaluation metrics, and the inherent challenges (*e.g.*, diversity, coherence, and realism for generation *v.s.* relevance scores in retrieval). To our best knowledge, previous works of DiffusionRet [Jin et al., 2023] and MomentDiff [Li et al., 2023a] combines Diffusion model with retrieval. Albeit the encouraging performance, their method adopts the Diffusion model to learn a mapping from random noise to the signal (e.g., similarity score or real moments) using the diffusion $L_1$ or $L_2$ loss, treating the ranking problem as a generation task without addressing the problem studied in this work.

## 3  Method

### 3.1  Preliminaries

**Text-Video Retrieval**. We have a multi-modality dataset consisting of $N_v$ video clips and $N_t$ text descriptions, *e.g.*, $\mathcal{D} = \{v^{(i)}, t^{(j)}\}$, where $i = 1, ..., N_v$, $j = 1, ..., N_t$. Each video clip $v^{(i)}$ may correspond to one or more text samples $t^{(j)}$. For simplicity, we let $v$ and $t$ to denote arbitrary video and text. The retrieval system adopts a multi-modality model, such as upon CLIP [Radford et al., 2021], to abstract the video and text features into a joint embedding space, *i.e.*, $\mathbf{v}, \mathbf{t} \in \mathbb{R}^d$, where $d$ denotes the embedding dimension of the joint space.

$$\mathbf{v} = F_\theta(v); \quad \mathbf{t} = G_\phi(t), \tag{1}$$

where $F_\theta(\cdot)$ denotes the advanced visual encoder and $G_\phi(\cdot)$ denotes the text encoder. Existing works mainly adopt the symmetric-formed cross-entropy [Oord et al., 2018] loss to train the model

$$\mathcal{L}_{\text{sce}} = -\frac{1}{B}\sum_{i=1}^{B}\log\frac{e^{s(\mathbf{t}^{(i)},\mathbf{v}^{(i)})\cdot\tau}}{\sum_j e^{s(\mathbf{t}^{(i)},\mathbf{v}^{(j)})\cdot\tau}} + \log\frac{e^{s(\mathbf{t}^{(i)},\mathbf{v}^{(i)})\cdot\tau}}{\sum_j e^{s(\mathbf{t}^{(j)},\mathbf{v}^{(i)})\cdot\tau}}, \tag{2}$$

where the cosine similarity $s(\mathbf{t}, \mathbf{v}) = \frac{|\mathbf{t}\cdot\mathbf{v}|}{|\mathbf{t}|\cdot|\mathbf{v}|}$ is employed as a distance measuring function and $B$ denotes the batch size. A learnable temperature scaling factor $\tau$ is employed to control the smoothness of the loss. The first term in Eq. (2) guides the text-to-video ($t \to v$) retrieval and the second term focuses on the video-to-text ($v \to t$) retrieval. Notably, the scenario of $\mathcal{L}_{\text{sce}} = 0$ presents a perfect alignment for all video and text embedding in the joint space, where the relevant $\frac{\mathbf{v}}{|\mathbf{v}|}$ and $\frac{\mathbf{t}}{|\mathbf{t}|}$ overlaps ($s(\mathbf{t}, \mathbf{v}) = 1$), and the irrelevant text-video are orthogonal, $i.e.$, $s(\mathbf{t}, \mathbf{v}) = 0$. While in practice, it is hard to achieve a perfect alignment considering the substantial disparities between the $t$ and $v$ data. To this end, a modality gap $\delta$ inevitably exists between the relevant text and video embedding as shown by Fig. 1 (a), which poses challenges in determining the relevancy. Nevertheless, how to model the modality gap $\delta$ and perform alignment accordingly can benefit the retrieval but does not seem to have been fully explored. Note that there are two ways to define the $\delta$ by either stemming from $\mathbf{t}$ or $\mathbf{v}$. Considering the one-to-many correspondence of video-to-text, as exampled in Fig. 2 ($\mathbf{v}^{(1)}$ $v.s.$ $\{\mathbf{t}^{(1)}, \mathbf{t}^{(2)}, \mathbf{t}^{(3)}\}$), each $\mathbf{t}$ associates to a unique $\delta$ but $\mathbf{v}$ corresponds to multiple $\delta$ vectors. Accordingly, we consider $\delta$ stemming from $\mathbf{t}$ and consistently adopt $\delta = \mathbf{v} - \mathbf{t}$ in this work.

**Fixed Prior Alignment**. We first posit a fixed prior for the multi-modality gap, similar to existing multi-modality alignment practices [Qiu et al., 2024, Nukrai et al., 2022], and have

$$\delta \sim \mathcal{N}(0, \sigma^2 I), \tag{3}$$

where $\sigma^2$ determines the scale of the modality gap, $i.e.$, to what extant $\mathbf{v}$ deviates from the relevant $\mathbf{t}$. Given the arbitrary text embedding $\mathbf{t}$, we draw $\delta$ from the fixed prior as shown above and impose it onto the embedding as the alignment, $e.g.$, $\mathbf{t} + \delta$. Then this aligned embedding will take effect in the retrieval system as a substitution of $\mathbf{t}$. Unfortunately, there is a decrease in retrieval performances for variant priors (see Table 4). To elaborate, the fixed prior is far from enough to capture the delicate distribution of $\delta$ given the complicated structure of the CLIP embedding space [Paszke et al., 2019]. Besides, the fixed prior is inflexible to adapt to varied text and video inputs, causing misalignment. These limitations naturally prompt us to resort to more advanced technologies for the gap modeling. As known, Diffusion models [Yang et al., 2023, Croitoru et al., 2023] approximate the sophisticated distributions without explicitly presuming a prior and has been proven to excel in bridging multi-modality representations [Zhou et al., 2023]. In this work, we investigate the potential of the Diffusion models for the gap modeling under text-video retrieval.

## 3.2 Proposed Diffusion-based Alignment

We define the target distribution of the Diffusion model upon the modality gap $\delta$. Following the previous works [Gorti et al., 2022, Guan et al., 2023], our proposed Diffusion model operates in the embedding space of the CLIP model and defines a Markov chain of the latent variables for $\delta$, $i.e.$, $\delta_1, ..., \delta_T$, where the initial state of the Diffusion model remains an isotropic Gaussian, $i.e.$, $\delta_T \sim \mathcal{N}(\mathbf{0}, \mathbf{I})$, which in this case, can also be treated as a fixed prior for the modality gap $\delta$, and $\delta_0$ corresponds to the ground truth modality gap. It is non-trivial incorporating the proposed Diffusion model into existing retrieval pipeline due to different natures of the generation and retrieval. To achieve this, we reimplement the denoising network, develop a pretraining stage and a reverse process-engaged retrieval pipeline.

**Denoising Network Design**. We first design a denoising network by referring to the prevalent DiT [Peebles and Xie, 2023]. The denoising network, denoted as $\epsilon_\gamma(\cdot)$, consists of $N$ DiT blocks and let $\gamma$ denote the learnable parameters for the network. As shown in Fig. 2, the denoising network takes three inputs, including the timestamp $t$, the latent embedding $\delta_t \in \mathbb{R}^d$, and the condition $\mathbf{c} \in \mathbb{R}^d$. Since we define the modality gap $\delta$ stems from $\mathbf{t}$ as in Section 3.1, we let the text embedding $\mathbf{t}$ as the condition to guide the generation, $i.e.$, $\mathbf{c} = \mathbf{t}$, which substitutes the vanilla classifier-free guidance [Ho and Salimans, 2021] in DiT. Finally, since we still perform the retrieval in the same latent space, there is no need to perform a reshape operation at the end of the denoising network. As shown in Fig. 1 (c), the denoising network $\epsilon_\gamma(\cdot)$ approximates the noise imposed on $\delta_t$ during the diffusion process. To facilitate the learning of $\epsilon_\gamma(\cdot)$, we introduce a pretraining stage as follows.

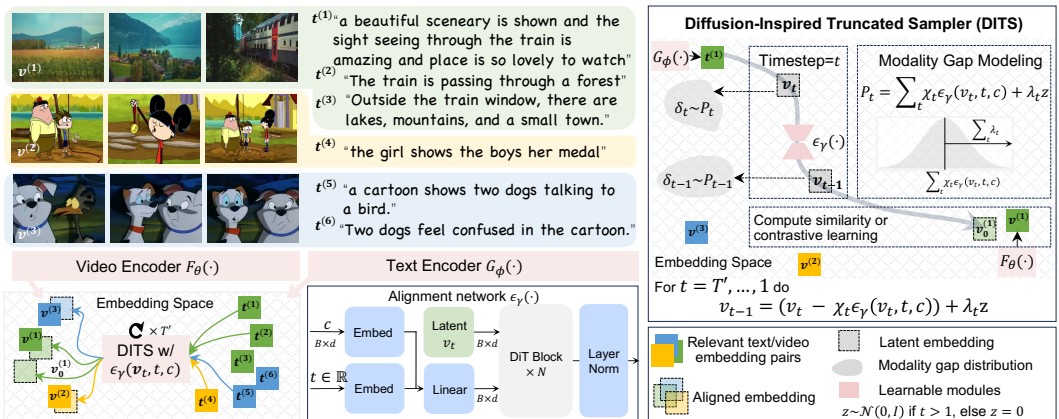

Figure 2: Overview of DITS. Given the video embedding $\mathbf{v}$ from $F_\theta(\cdot)$ and the text embedding $\mathbf{t}$ from $G_\phi(\cdot)$, DITS performs a $T'$-steps sampling by starting from $\mathbf{t}$, to gradually approach $\mathbf{v}$ (as exampled by a relevant pair $\mathbf{t}^{(1)}, \mathbf{v}^{(1)}$) with the aligned embedding $\mathbf{v}_0^{(1)}$ at the timestamp $t = 0$. Meanwhile the progressive alignment, DITS intrinsically approximates the distribution of the modality gap $\delta \sim P_t$ at each timestamp $t = T', ..., 1$. The contrastive loss is adopted to guide the alignment and modeling, with parameters: $\Theta = \{\theta, \phi, \gamma\}$. We devise $\epsilon_\gamma(\cdot)$ upon DiT [Peebles and Xie, 2023].

**Diffusion model Pretraining.** The target distribution of $\delta$ can be highly adaptive to $\mathbf{v}$ and $\mathbf{t}$ given by Eq. (1). To stabilize the learning, we adopt the pretrained retrieval model and fix $F_\theta(\cdot)$ and $G_\phi(\cdot)$ during the pretraining. The learning objective of $\epsilon_\gamma(\cdot)$ derives from the diffusion process [Qiu et al., 2024, Nukrai et al., 2022], which gradually adds the Gaussian noise to $\delta_0$ for $T$ steps, where $\delta_1, ..., \delta_T$ are the latent variables of the same dimension, *i.e.*, $\delta_t \in \mathbb{R}^d$

$$q(\delta_{1:T}|\delta_0) := \prod_{t=1}^{T} q(\delta_t|\delta_{t-1}), \quad q(\delta_t|\delta_{t-1}) := \mathcal{N}(\delta_t; \sqrt{1-\beta_t}\delta_{t-1}, \beta_t\mathbf{I}), \tag{4}$$

where $\beta_1, ..., \beta_T$ can simply be held as constant [Ho et al., 2019] to form a variance schedule. By adopting the reparameterization [Kingma and Welling, 2013] upon Eq. (4), $\delta_t$ can be analytically represented in a closed form

$$q(\delta_t|\delta_0) = \mathcal{N}(\delta_t; \sqrt{\bar{\alpha}_t}\delta_0, (1-\bar{\alpha}_t)\mathbf{I}), \quad \text{where} \quad \alpha_t := 1 - \beta_t, \quad \bar{\alpha}_t := \prod_{s=1}^{t} \alpha_s. \tag{5}$$

To approximate $P_\gamma(\delta_{t-1}|\delta_t)$, we use the variational lower bound to optimize the negative log-likelihood $-\log P_\gamma(\delta_0)$, which can be parameterized as a $\mathcal{L}_2$ loss [Ho et al., 2019]

$$\mathcal{L}_\gamma = \mathbb{E}_{\delta_0, t, \epsilon}[||\epsilon - \epsilon_\gamma(\sqrt{\bar{\alpha}}\delta_0 + \sqrt{1-\bar{\alpha}_t}\epsilon, t, \mathbf{c})||^2], \tag{6}$$

where $\epsilon \sim \mathcal{N}(\mathbf{0}, \mathbf{I})$, $t \sim \text{Uniform}(1, ..., T)$ and $\mathbf{c}$ denotes the condition. Specifically, $\epsilon$, $t$, and $\mathbf{c}$ are three inputs of the denoising network $\epsilon_\gamma(\cdot)$. By minimizing the $\mathcal{L}_\gamma$, the Diffusion model learns to transition from the isotropic Gaussian to the target distribution of $\delta_0$ through a reverse process, based on which this method aligns the input text embedding $\mathbf{t}$ (given as condition) as $\mathbf{t} + \delta_0$, to bridge its relevant counterpart $\mathbf{v}$. We introduce the reverse process-engaged retrieval pipeline in the next.

**Reverse Process-Engaged Retrieval.** We perform the vanilla DDPM [Ho et al., 2019] sampling process in this work. Starting from the isotropic Gaussian $\delta_T \sim \mathcal{N}(\mathbf{0}, \mathbf{I})$, the reverse process samples the gap $\delta_0$ through a $T$-step Markov chain

$$\delta_{t-1} = \frac{1}{\sqrt{\alpha_t}}(\delta_t - \frac{1-\alpha_t}{\sqrt{1-\bar{\alpha}_t}}\epsilon_\gamma(\delta, t, \mathbf{c})) + \sigma_t\mathbf{z}, \quad \text{where} \quad \mathbf{z} \sim \mathcal{N}(\mathbf{0}, \mathbf{I}), \tag{7}$$

where $t$ decreases from $T$ to 1, $\mathbf{z} = \mathbf{0}$ if $t = 1$, and $\sigma_t^2 = \beta_t$. The reverse process of the Diffusion model will be plugged into the retrieval pipeline as the post-feature extraction alignment, collectively forming a reverse process-engaged retrieval pipeline. To this end, one can substitute $\mathbf{t}$ with the aligned embedding $\mathbf{t} + \delta_0$ and perform retrieval with regard to arbitrary $\mathbf{v}$. Since the embedding space structure has been edited by the Diffusion model, it might be better to further rearrange the multi-modality embeddings with the contrastive loss. We opt to jointly tune all of the learnable parameters $\Theta = \{\theta, \gamma, \phi\}$ upon this reverse process-engaged retrieval pipeline with the symmetric cross-entropy given in Eq. (2). We provide more details and discussion in Section 4.1.

### 3.3 Proposed Alignment by DITS

To sum up, this work first implements a feasible approach to leverage the generative power of the Diffusion model in the retrieval pipeline. We firstly pretrain the Diffusion model with Eq. (6) and then incorporate the reverse process in Eq. (7) for the alignment. Notably, we find that despite that the Diffusion model effectively bridging the relevant text and video embedding, either applying the pretrained model for the alignment or adopting the reverse process-engaged retrieval causes obvious performance declines (shown in Table 4). This prompt us to rethink Diffusion models in the context of our problem. Based on the designs in Section 3.2, we develop DITS in the following.

**Rethinking the Vanilla Diffusion model**. We find two-fold limitations of the Diffusion model defective for the task of retrieval. **First**, the $\mathcal{L}_2$ loss ($\mathcal{L}_\gamma$) does not fit the ranking problem inherent in text-video retrieval. On one hand, $\mathcal{L}_\gamma$ shown in Eq. (6) minimizes a mean squared error, which inherently lacks an prioritization effect for the relevant pairs over irrelevant ones. On the other, it is not clear what the desired alignment should be for the irrelevant pairs, making it hard to define a target distribution accordingly and implement $\mathcal{L}_\gamma$ for the modeling. Nevertheless, there is no guarantee that the $\mathcal{L}_2$ loss (*e.g.*, $\mathcal{L}_\gamma$) can effectively pull away irrelevant pairs, even though the gap is mitigated for the relevant pairs as shown in Fig. 3. **Second**, the reverse process either fails to benefit the retrieval (Table 4) – the initial starting point of the reverse, *i.e.*, of $\delta_T \sim \mathcal{N}(\mathbf{0}, \mathbf{I})$

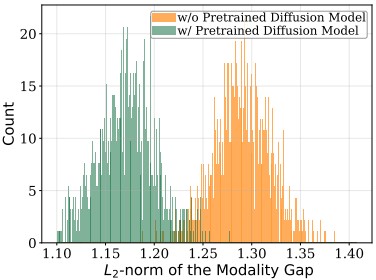

Figure 3: Modality gap distribution for the **relevant** pairs before and after the Diffusion model alignment.

can significantly impact the diversity and the generation quality [Zhou et al., 2023, Chung et al., 2022, Zheng et al., 2023]. This can incur inaccurate alignment and retrieval. How to determine the $\delta_T$ without affecting the performance remains a challenge.

**Diffusion-Inspired Truncated Sampler**. Bearing the above findings, in this work, we introduce the diffusion-inspired truncated sampler (DITS) as an effective alignment tool to benefit the retrieval. Considering the neighboring nature of the CLIP embedding space – relevant $\mathbf{v}$ and $\mathbf{t}$ have already been within arm's length of the distance) [Liang et al., 2022, Nukrai et al., 2022] – there is no need to start sampling from the isotropic Gaussian as in the vanilla Diffusion models. Instead, DITS proposes a progressive alignment procedure that directly starts from the $\mathbf{t}$ to $\mathbf{v}$ as shown in Fig. 2 *upper right*, forming truncation. Given $\mathbf{v}$ by $F_\theta(\cdot)$ and $\mathbf{t}$ by $G_\phi(\cdot)$

$$\mathbf{v}_{t-1} = (\mathbf{v}_t - \chi_t \epsilon_\gamma(\mathbf{v}_t, t, \mathbf{c})) + \lambda_t \mathbf{z}, \ \text{For} \ t = T', ..., 1, \tag{8}$$

where $\epsilon_\gamma(\cdot)$ serves as an alignment network and $T'$ denotes the number of the truncated timestamps. The starting point $\mathbf{v}_{T'} = \mathbf{t}$ (taking $\mathbf{t}^{(1)}$ as an example in Fig. 2) and $\chi_t = \frac{1-\alpha_t}{\sqrt{\alpha_t - \alpha_t \bar{\alpha}_t}}$ denotes a pre-defined schedule. Besides, we let $\mathbf{z} \sim \mathcal{N}(\mathbf{0}, \mathbf{I})$ if $t > 1$, else $\mathbf{z} = \mathbf{0}$. We set $\lambda_t = \sigma_t$. Compared with vanilla sampling process given by Eq. (7), the initial latent of DITS becomes a text embedding point (*e.g.*, $\mathbf{t}^{(i)}$) rather than a random sample from the isotropic Gaussian, which brings following advantages: (1) DITS gets rid of the effect of starting point random sampling. (2) the query text (such as for $t \to v$) initializes the alignment in a more straightforward way, rather than solely being a condition to guide the learning. Considering this, the condition $\mathbf{c}$ is not used in $\epsilon_\gamma(\cdot)$. In addition, we adopt the contrastive loss upon Eq. (2) to guide the learning, during which we substitute $\mathbf{t}$ with the aligned embedding $\mathbf{v}_0$. We jointly train all of the learnable parameters $\Theta = \{\theta, \phi, \gamma\}$ as denoted in red in Fig. 2. By observing and considering both relevant and irrelevant pairs, DITS learns the alignment that potentially benefits the overall retrieval performance.

It turns out DITS intrinsically involves the modality gap modeling along the timestamps, (*i.e.*, $\delta \sim P_t$) as shown in Fig. 2. One can derive the distribution of the modality gap $\delta$ from Eq. (8), which can be represented by using the reparameterization

$$P_t = \sum_t \chi_t \epsilon_\gamma(\mathbf{v}_t, t, \mathbf{c}) + \lambda_t \mathbf{z}, \tag{9}$$

where $\epsilon_\gamma(\mathbf{v}_t, t, \mathbf{c})$ can be regarded as a bias of the modality gap and $\lambda_t$ controls the scale of the gap modeling at the timestamp $t$. Interestingly, DITS yields a remarkable performance boost over the baseline and achieves the state-of-the-art performance (Section 4.2). We also empirically find that DITS also serves as an effective tool to help improve the CLIP embedding space. See more details in Section 4.4.

Table 1: Text-to-video retrieval performance on MSRVTT [Xu et al., 2016] and LSMDC [Rohrbach et al., 2015]. Bold denotes the best performance. "–" denotes that the result is unavailable.

| Methods | MSRVTT Retrieval | | | | | LSMDC Retrieval | | | | |
|---|---|---|---|---|---|---|---|---|---|---|
| | R@1↑ | R@5↑ | R@10↑ | MdR↓ | MnR↓ | R@1↑ | R@5↑ | R@10↑ | MdR↓ | MnR↓ |
| **CLIP-ViT-B/32** | | | | | | | | | | |
| X-Pool [Gorti et al., 2022] | 46.9 | 72.8 | 82.2 | 2.0 | 14.3 | 25.2 | 43.7 | 53.5 | 8.0 | 53.2 |
| STAN [Liu et al., 2023] | 46.9 | 72.8 | 82.8 | 2.0 | – | 23.7 | 42.7 | 51.8 | 9.0 | – |
| ProST [Li et al., 2023b] | 48.2 | 74.6 | 83.4 | 2.0 | 12.4 | 24.1 | 42.5 | 51.6 | 9.0 | 54.6 |
| DiffusionRet [Jin et al., 2023] | 49.0 | 75.2 | 82.7 | 2.0 | 12.1 | 24.4 | 43.1 | 54.3 | 8.0 | **40.7** |
| UATVR [Fang et al., 2023] | 47.5 | 73.9 | 83.5 | 2.0 | 12.3 | – | – | – | – | – |
| UCOFIA [Wang et al., 2023] | 49.4 | 72.1 | – | – | 12.9 | – | – | – | – | – |
| TEFAL [Ibrahimi et al., 2023] | 49.4 | **75.9** | 83.9 | 2.0 | 12.0 | 26.8 | 46.1 | 56.5 | 7.0 | 44.4 |
| CLIP-ViP [Xue et al., 2023] | 50.1 | 74.8 | 84.6 | 1.0 | – | 25.6 | 45.3 | 54.4 | 8.0 | – |
| T-MASS [Wang et al., 2024a] | 50.2 | 75.3 | **85.1** | 1.0 | 11.9 | **28.9** | **48.2** | **57.6** | 6.0 | 43.3 |
| DITS (Ours) | **51.9** | 75.7 | 84.6 | **1.0** | **11.6** | 28.2 | 47.3 | 56.6 | **6.0** | 43.7 |
| **CLIP-ViT-B/16** | | | | | | | | | | |
| X-Pool [Gorti et al., 2022] | 48.2 | 73.7 | 82.6 | 2.0 | 12.7 | 26.1 | 46.8 | 56.7 | 7.0 | 47.3 |
| UATVR [Fang et al., 2023] | 50.8 | 76.3 | 85.5 | 1.0 | 12.4 | – | – | – | – | – |
| CLIP-ViP [Xue et al., 2023] | 54.2 | 77.2 | 84.8 | 1.0 | – | 29.4 | 50.6 | 59.0 | 5.0 | – |
| T-MASS [Wang et al., 2024a] | 52.7 | 77.1 | 85.6 | 1.0 | 10.5 | 30.3 | 52.2 | **61.3** | 5.0 | 40.1 |
| DITS (Ours) | **55.0** | **79.8** | **87.1** | **1.0** | **10.0** | **31.0** | **52.4** | 61.0 | **5.0** | **38.4** |

Table 2: Text-to-video retrieval performance on DiDeMo [Anne Hendricks et al., 2017] and VA-TEX [Wang et al., 2019]. Bold denotes the best performance. "–" denotes that the result is unavailable.

| Methods | DiDeMo Retrieval | | | | | VATEX Retrieval | | | | |
|---|---|---|---|---|---|---|---|---|---|---|
| | R@1↑ | R@5↑ | R@10↑ | MdR↓ | MnR↓ | R@1↑ | R@5↑ | R@10↑ | MdR↓ | MnR↓ |
| **CLIP-ViT-B/32** | | | | | | | | | | |
| X-Pool [Gorti et al., 2022] | 44.6 | 73.2 | 82.0 | 2.0 | 15.4 | 60.0 | 90.0 | 95.0 | 1.0 | 3.8 |
| ProST [Li et al., 2023b] | 44.9 | 72.7 | 82.7 | 2.0 | 13.7 | 60.6 | 90.5 | 95.4 | 1.0 | 3.4 |
| STAN [Liu et al., 2023] | 46.5 | 71.5 | 80.9 | 2.0 | – | – | – | – | – | – |
| DiffusionRet [Jin et al., 2023] | 46.7 | 74.7 | 82.7 | 2.0 | 14.3 | – | – | – | – | – |
| UATVR [Fang et al., 2023] | 43.1 | 71.8 | 82.3 | 2.0 | 15.1 | 61.3 | 91.0 | 95.6 | 1.0 | 3.3 |
| UCOFIA [Wang et al., 2023] | 46.5 | 74.8 | – | – | 13.4 | 61.1 | 90.5 | – | – | 3.4 |
| CLIP-ViP [Xue et al., 2023] | 48.6 | 77.1 | 84.4 | 2.0 | – | – | – | – | – | – |
| T-MASS [Wang et al., 2024a] | 50.9 | 77.2 | 85.3 | 1.0 | 12.1 | 63.0 | 92.3 | 96.4 | 1.0 | 3.2 |
| DITS (Ours) | **51.1** | **77.9** | **85.8** | **1.0** | 12.1 | **64.1** | **92.7** | **97.0** | **1.0** | **2.9** |
| **CLIP-ViT-B/16** | | | | | | | | | | |
| X-Pool [Gorti et al., 2022] | 47.3 | 74.8 | 82.8 | 2.0 | 14.2 | 62.6 | 91.7 | 96.0 | 1.0 | 3.4 |
| UATVR [Fang et al., 2023] | 45.8 | 73.7 | 83.3 | 2.0 | 13.5 | 64.5 | 92.6 | 96.8 | 1.0 | 2.8 |
| CLIP-ViP [Xue et al., 2023] | 50.5 | 78.4 | 87.1 | 1.0 | – | – | – | – | – | – |
| T-MASS [Wang et al., 2024a] | 53.3 | 80.1 | **87.7** | 1.0 | **9.8** | 65.6 | 93.9 | 97.2 | 1.0 | 2.7 |
| DITS (Ours) | **55.8** | **80.5** | 87.5 | **1.0** | 11.0 | **66.4** | **94.3** | **97.5** | **1.0** | **2.7** |

# 4 Experiment

## 4.1 Experimental Settings

**Datasets**. We employ five benchmark datasets for evaluation. Firstly, we utilize MSRVTT [Xu et al., 2016], comprising $10,000$ YouTube video clips (each having 20 captions) and follow the 1K-A testing split in Liu et al. [2019]. Secondly, LSMDC [Rohrbach et al., 2015] includes $118,081$ text-video pairs, providing videos with longer duration. The testing set contains 1000 videos, as per Gabeur et al. [2020], Gorti et al. [2022]. Thirdly, DiDeMo [Anne Hendricks et al., 2017] contains $\sim 40,000$ captions and $\sim 10,000$ video clips. We adhere to the data splits detailed in Luo et al. [2022], Jin et al. [2023]. Fourthly, VATEX [Wang et al., 2019] comprises $41,250$ video clips, where each is paired with ten English and ten Chinese descriptions. We follow the split in Chen et al. [2020a]. Lastly, Charades [Sigurdsson et al., 2016] contains 9848 video clips, each with multiple text descriptions detailing daily activities and actions. We adopt the split protocol of Lin et al. [2022].

**Implementation Details**. Following previous works [Gorti et al., 2022, Li et al., 2023b], we resize the video to be the spatial size of $224 \times 224$ and uniformly sample 12 frames from the video for all datasets. For the retrieval model, we adopt X-Pool [Gorti et al., 2022] as the baseline, where both CLIP-ViT/B-32 and CLIP-VIT/B-16 are employed as the feature extractor. The dropout is set to $0.3$. Different from DiT [Peebles and Xie, 2023], we set our denoising network (also used as the alignment network in DITS) with $N = 4$ blocks, with 16 heads and an MLP ratio of $4.0$. We let the dimension $d = 512$ for the whole model. We find that a timestamp of $T = 10$ is enough for diffusion-based alignment. For DITS, we set the truncated timestamp $T' = 5$ for DiDeMo and $T' = 10$ for others. A linear variance schedule with $\beta = 0.1$ and $\beta = 0.99$ is adopted. All of the parameters $\Theta = \{\theta, \phi, \gamma\}$ are trained with an AdamW [Loshchilov and Hutter, 2017] optimizer with weight decay of $0.2$ and

Table 3: Text-to-video comparisons on Charades [Sigurdsson et al., 2016]. Bold denotes the best.

| Methods | R@1 | R@5 | R@10 | MdR | MnR |
|---|---|---|---|---|---|
| *CLIP-ViT-B/32* | | | | | |
| ClipBERT [Lei et al., 2021] | 6.7 | 17.3 | 25.2 | 32.0 | 149.7 |
| CLIP4Clip [Luo et al., 2022] | 9.9 | 27.1 | 36.8 | 21.0 | 85.4 |
| X-Pool [Gorti et al., 2022] | 11.2 | 28.3 | 38.8 | 20.0 | 82.7 |
| T-MASS [Wang et al., 2024a] | 14.2 | 36.2 | **48.3** | 12.0 | **54.8** |
| DITS (Ours) | **15.5** | **36.5** | 47.8 | **12.0** | 55.0 |
| *CLIP-ViT-B/16* | | | | | |
| CLIP4Clip [Luo et al., 2022] | 16.0 | 38.2 | 48.5 | 12.0 | 54.1 |
| X-Pool [Gorti et al., 2022] | 20.7 | 42.5 | 53.5 | 9.0 | 47.4 |
| T-MASS [Wang et al., 2024a] | 26.7 | 51.7 | 63.9 | 5.0 | 30.0 |
| DITS (Ours) | **27.4** | **52.0** | **64.3** | **5.0** | **29.7** |

Table 4: Ablation study of the proposed DITS on MSRVTT-1k. We adopt X-Pool [Gorti et al., 2022] as the "Baseline". "Diffusion" denotes the Diffusion-based alignment method in Section 3.2.

| Methods | | Alignment | $\mathcal{L}_2$ | $\mathcal{L}_{\mathrm{sce}}$ | Truncation | R@1 | R@5 | R@10 | MdR | MnR |
|---|---|---|---|---|---|---|---|---|---|---|
| Baseline | | ✗ | ✗ | ✓ | ✗ | 46.9 | 72.8 | 82.2 | 2.0 | 14.3 |
| Fixed Prior | $\sigma^2 = 1.0$ | ✓ | ✗ | ✓ | ✗ | 28.0 | 61.9 | 74.7 | 3.0 | 19.7 |
| | $\sigma^2 = 0.1$ | ✓ | ✗ | ✓ | ✗ | 45.1 | 73.3 | 82.6 | 2.0 | 13.8 |
| Diffusion | Pretrain | ✓ | ✓ | ✗ | ✗ | 35.4 | 66.7 | 78.3 | 3.0 | 14.4 |
| | Fine tune | ✓ | ✗ | ✓ | ✗ | 46.5 | 73.9 | 83.0 | 2.0 | 13.3 |
| DITS | | ✓ | ✗ | ✓ | ✓ | **51.9** | **75.7** | **84.6** | **1.0** | **11.6** |

warmup rate of $0.1$. We set the training epochs to $5$ for all datasets and adopt the same seed of $24$. We perform contrastive learning with a batch size of $B = 32$ for all datasets and backbones. Same as X-Pool [Gorti et al., 2022], the learning rate of the CLIP model is initialized as $1 \times 10^{-5}$. The learning rate for non-CLIP modules is $3 \times 10^{-5}$ for MSRVTT [Xu et al., 2016] and $1 \times 10^{-5}$ for all the other datasets. For a fair comparison, no post-processing techniques [Bogolin et al., 2022, Cheng et al., 2021] are employed in this work. We implement DITS with PyTorch [Paszke et al., 2019] and perform experiments on an NVIDIA A100 GPU.

**Compared Methods**. We compare DITS with X-Pool [Gorti et al., 2022], STAN [Liu et al., 2023], ProST [Li et al., 2023b], DiffusionRet [Jin et al., 2023], UATVR [Fang et al., 2023], UCOFIA [Wang et al., 2023], TEFAL [Ibrahimi et al., 2023], CLIP-ViP [Xue et al., 2023] and T-MASS [Wang et al., 2024a]. Among them, TEFAL adopts additional audio data to train the model. CLIP-ViP further adopts WebVid-2.5M [Bain et al., 2021] and HD-VILA-100M [Xue et al., 2022] as the augmentation. Besides, some methods take larger batch sizes for the training (*e.g.*, $64$ in DiffusionRet [Jin et al., 2023] and UATVR, $128$ for ProST, STAN, and UCOFIA except DiDeMo). We perform evaluation using metrics such as Recall at ranks $\{1, 5, 10\}$, Median Rank (MdR), and Mean Rank (MnR).

### 4.2 Performance Comparison

We compare the text-to-video retrieval performance of different methods in Table 1, 2, and 3. The proposed DITS can achieve the best performance on different CLIP backbones, datasets, and metrics. For example, on MSRVTT, DITS outperforms the baseline X-Pool by $5\%$ at R@1 and outperforms T-MASS by $1.7\%/2.3\%$ at R@1 on different backbones. On DiDeMo, DITS outperforms CLIP-ViT by $2.2\%$ with CLIP-ViT-B/32 and $5.3\%$ with CLIP-ViT-B/16, achieving a consistent boost on different metrics. Moreover, on the challenging dataset of Charades, DITS also brings a remarkable performance boost over X-Pool on both backbones, *e.g.*, $4.3\%$ and $6.7\%$ at R@1, respectively. There is one scenario that T-MASS enables better retrieval than DITS on LSMDC with CLIP-ViT-B/32. However, our method gains an advantage with a larger CLIP backbone on all datasets, indicating better scalability on the retrieval model. We also note that DITS excels in retrieval top-ranked results, such as on MSRVTT, DiDeMo – a progressive alignment and gap modeling facilitates a narrower retrieval scope, encouraging retrieval precision. We provide in-depth analysis of this tendency in Section 4.4. Overall, extensive results on benchmark datasets demonstrate that DITS effectively aligns text-video embedding in the joint space, delivering promising retrieval results.

### 4.3 Ablation Study

Table 4 provides an ablation study of the proposed method. Specifically, "Baseline" denotes the baseline method of X-Pool [Gorti et al., 2022], based on which we provide three types of the

Table 5: Discussion on DITS. Highlighted settings are adopted for the benchmark comparison.

(a) Discussion on the timestamps ($T'$) on MSRVTT.

| $T'$ | R@1 ↑ | R@5 ↑ | R@10 ↑ | MdR ↓ | MnR ↓ |
|------|-------|-------|--------|-------|-------|
| 0 | 46.9 | 72.8 | 82.2 | 2.0 | 14.3 |
| 1 | 49.2 | 75.0 | **85.1** | 2.0 | **11.3** |
| 10 | **51.9** | **75.7** | 84.6 | **1.0** | 11.6 |
| 20 | 47.0 | 73.9 | 83.7 | 2.0 | 12.2 |
| 30 | 32.5 | 63.5 | 76.0 | 3.0 | 16.2 |
| 40 | 0.2 | 0.5 | 1.0 | 493.5 | 497.9 |

(b) The effect of DITS on CLIP. We study the DITS alignment in the fixed CLIP space and upon a learnable CLIP model. By aligning the embedding, DITS guides the CLIP learning and improves the space.

| Methods | R@1 | R@5 | R@10 | MdR | MnR |
|---------|-----|-----|------|-----|-----|
| Baseline w/o DITS | 46.9 | 72.8 | 82.2 | 2.0 | 14.3 |
| DITS (Fix CLIP) | 39.2 | 65.4 | 77.1 | 2.0 | 18.0 |
| DITS (Joint train) | **51.9** | **75.7** | **84.6** | **1.0** | **11.6** |

alignment methods, corresponding to Fig. 1. First, we show the naive modeling methods using two different fixed priors, such as $\sigma^2 = 0.1, 1.0$. Both of them bring performance descent. Fixed priors are inflexible, hardly benefiting the retrieval as expected. We notice that the sensitivity of the performance toward the prior. This also prompts us to leverage advanced methods of Diffusion model below. We provide the retrieval performance corresponding to both training stages of pretraining and fine-tuning as in Section 3.2. In pretraining, the reverse process is adopted for the generation. As shown, the pretraining with $\mathcal{L}_2$ loss undermines the retrieval performance (*e.g.*, $35.4\%$ *v.s.* $46.9\%$). Despite the fact that $\mathcal{L}_2$ loss effectively attracts relevant pairs (Fig. 3), it fails to align the irrelevant ones, leading to the performance descent. By comparison, when we fine-tune the model with the contrastive loss, we observe a remarkable boost of $11.1\%$ at R@1 over pretraining and better results than baseline of X-Pool on metrics except R@1. This indicates that such a method, although enables a good retrieval scope (*e.g.*, being favorable for R@5/10), but fails to performs accurate alignment and retrieval, underlying which, one main flaw lies in the random sampling in isotropic Gaussian. Drawing inspiration, the proposed DITS in the bottom line of Table 4 proposes a truncation process to alleviate the above concern, and adopts contrastive learning to guide the alignment. Encouragingly, we find an evident boost of implementing such a pipeline by following the variance schedule of the vanilla diffusion flow (*e.g.*, $\alpha_t$, $\beta_t$). This prompt us to study the behavior of DITS in the following.

## 4.4 Discussion on DITS

**Discussion on Truncated timestamps**. The number of the timestamps $T'$ plays an important role for an accurate retrieval. We provide the performance of DITS at different timestamps in Table 5a. The $T' = 0$ denotes the baseline of X-Pool. As shown, the performance of DITS first increases and then drops when we gradually enlarging $T'$ from 1 to 40. DITS achieves the best performance at $T' = 10$. Specifically, one-step operation ($T' = 1$) enables determining a rough retrieval scope, benefiting Recall over a larger range, *e.g.*, R@10. Based on this, DITS requires more steps to narrow the scope for more accurate alignment and retrieval (*e.g.*, $T' = 10$). However, continually enlarging $T'$ enforces DITS to learn an extremely precise result, sacrificing the flexibility to accommodate the testing data and in the end, resulting the loss of the generalization ability.

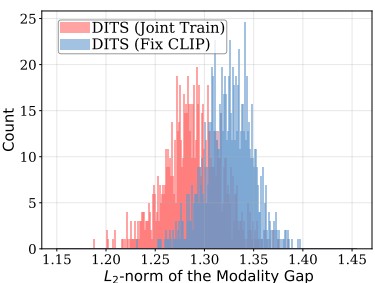

Figure 4: DITS reduces the modality gap for the **relevant** pairs, serving as a tool to improve the space by guiding the CLIP's learning.

**Improving the CLIP Embedding Space**. We also study the effect of DITS to the CLIP in Table 5b. There is a large performance gap when we fix CLIP or perform joint train for DITS. This indicates that solely performing the alignment in the original CLIP space is not enough. To this end, DITS not only plays the role of text-video embedding alignment, but also serves as a tool to guide the learning of CLIP, thus improving the CLIP space. As shown in Fig. 4, the distribution of the modality gap[3] between two spaces deviates from each other, characterizing the aligning effect of DITS.

**Diffusion Model Conditions**. We perform different conditions of the $\epsilon_\gamma(\cdot)$ in DITS, including with text condition **t**, with video condition **v**, and with both conditions **t** and **v**, as well as without any conditions (see discussions in Section 3.3). As shown in Table 6a, we find that there is no need to introduce the conditions onto the network. The diffusion process for this task emphasizes "accurate alignment", unlike the general Diffusion models highlighting diversity in creative works. The previous intuition of the diffusion condition may not be applicable in this work: (1) Due to the one-to-many

---

[3]Measured by $L_2$-norm (non-bounded), corresponding to Euclidean distance. We also provide mean absolute distance version (using $L_1$-norm of the same data in Fig. 6).

Table 6: Discussion on DITS. Highlighted settings are adopted for the benchmark comparison.

(a) Different model conditions for DITS on MSRVTT.

| Condition | R@1 | R@5 | R@10 | MdR | MnR |
|---|---|---|---|---|---|
| w/o alignment | 46.9 | 72.8 | 82.2 | 2.0 | 14.3 |
| $\mathbf{t}$ | 42.3 | 70.6 | 80.9 | 2.0 | 14.3 |
| $\mathbf{v}$ | 23.1 | 51.9 | 65.9 | 5.0 | 24.8 |
| $\mathbf{t}, \mathbf{v}$ | 40.9 | 71.3 | 80.8 | 2.0 | 15.0 |
| w/o condition | **50.1** | **75.5** | **84.9** | **1.0** | **12.2** |

(b) Different types of the modality gap $\delta$ on MSRVTT.

| $\delta$ | R@1 $\uparrow$ | R@5 $\uparrow$ | R@10 $\uparrow$ | MdR $\downarrow$ | MnR $\downarrow$ |
|---|---|---|---|---|---|
| w/o alignment | 46.9 | 72.8 | 82.2 | 2.0 | 14.3 |
| $\delta = \mathbf{t} - \mathbf{v}$ | 39.2 | 65.4 | 77.1 | 2.0 | 18.0 |
| $\delta = \mathbf{v} - \mathbf{t}$ (Ours) | **50.1** | **75.5** | **84.9** | **1.0** | **12.2** |

mapping of video-to-text, one video can map to multiple gap vectors. Taking the video as a condition can guide the model learn the undesired modality gap, empirically dropping the performance. (2) DITS start from the text, ensuring the alignment starts from a semantically meaningful initial point. The text condition acts as the constraint or guideline for the learning. Simultaneously applying the text embedding as the starting point and the condition can cause conflicting instructions, empirically decreasing the performance. (3) Joint usage of text and video conditions can inherent both limitations, empirically leading to an unsatisfactory performance.

**Modality Gap Stemming from video embedding**. We also provide experiments when let $\delta = \mathbf{t} - \mathbf{v}$ in Table 6b. As shown, such a setting brings sub-optimal performance. Imposing $\delta = \mathbf{t} - \mathbf{v}$ to the $\mathbf{v}$ works like approximating $\mathbf{t}$ for the retrieval, conducting the "text-to-text" retrieval, considering less semantic clues within the text, it might be hard to retrieve accurately. By comparison, the proposed method DITS implements $\delta = \mathbf{v} - \mathbf{t}$ to the $\mathbf{t}$ works like approximating $\mathbf{v}$ for the retrieval, performing the "video-to-video" retrieval. Since video contains much abundant clues, it might be easier to get better performance. Interesting, we notice that the previous work of Cap4video [Wu et al., 2023] also demonstrates similar tendencies in Table 5, denoted as `Query-Caption/Video Only` case.

## 5 Conclusion

This work studied the task of text-video retrieval by proposing a Diffusion-Inspired Truncated Sampler (DITS) for the multi-modality alignment. The primary contribution of this work was to offer insights on tailoring the Diffusion model for the ranking task inherent in retrieval. We uncovered two-fold limitations of the vanilla Diffusion models: the vanilla $\mathcal{L}_2$ loss was inadequate for alignment by overlooking the irrelevant pairs, the the random initial sampling in isotropic Gaussian introduces variability, causing misalignment. We proposed DITS, which leveraged the inherent proximity of text and video embedding and directly started sampling from text to alleviate the sampling variability, and proposed to adopt the contrastive learning not only to guide the iterative alignment steps over time, but also to facilitate gap modeling. Extensive experiments demonstrated the state-of-the-art performance of DITS. We found DITS could encourage an improved embedding space by guiding the CLIP's learning. We hope DITS can inspire future explorations in studying Diffusion models in the task of text-video retrieval.

## 6 Acknowledgment

This research work was supported by the DEVCOM Army Research Laboratory under contract W911QX-21-D-0001.

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

# A    Appendix / supplemental material

We provide in-depth discussions, more results of the proposed DITS as follows

- Limitations discussion. (Section A.1).
- Broader impacts on the proposed method. (Section A.2).
- More discussions and results on Diffusion-based alignment (Section A.3)
- More discussions on computational cost and efficiency. (Section A.4).

## A.1    Limitations

Despite the advancements brought by the proposed Diffusion-Inspired Truncated Sampler (DITS), there might be some limitations can be considered for the future research.

DITS, like other multi-modality alignment methods using contrastive learning, requires redundant amounts of data to train effectively. In scenarios where labeled text-video pairs are scarce, the performance of DITS remains unexplored. This raises a question on studying DITS on challenging text-video retrieval scenarios or zero-shot, few-shot retrieval tasks.

In addition, the robustness of DITS to noisy or incomplete data may be a potential concern. In real-world scenarios, text descriptions and video content can be noisy, incomplete, or of poor quality. While DITS aims to bridge the modality gap, its performance may still degrade in the presence of such imperfect data. This highlights the need for further research into deploying DITS to real-world retrieval systems with data imperfections.

## A.2    Broader Impacts

The proposed Diffusion-Inspired Truncated Sampler (DITS) presents a new alignment solution in the realm of text-video retrieval. By effectively modeling and aligning the modality gap, DITS elevates the performance of retrieval systems and may contribute to the broader landscape of multi-modality learning and the diffusion models.

In essence, DITS offers a novel methodology for mitigating the disparities between textual and visual data, potentially inspiring diverse applications, for example, image captioning models that can accurately describe the contents of images, image question answering models that requires precise semantics understanding capabilities of the model toward the visual information.

Beyond its impact on the multi-modality learning, DITS presents a new diffusion flow that is guided by the contrastive learning and considers nature of the ranking task. In light of this, DITS can pave the way for future innovations in the diffusion model design for learning the relationships of the multi-modality data. Besides, traditional diffusion models often struggle with maintaining alignment accuracy due to their reliance on fixed priors and isotropic Gaussian initial states. DITS introduces a truncated diffusion process that shifts from a Gaussian noise start to a meaningful intermediate state , showcasing a new way to handle complex alignment tasks in diffusion models.

Despite the encouraging performance of the proposed method. There exists a possibility that the proposed method can be utilized in the real-world applications with the illegitimate purpose.

Table 7: Discussion on $\mathcal{L}_1$ Loss. We compare the retrieval performance with the pretrained Diffusion model as an alignment. We perform experiments on MSRVTT. All the other settings are kept the same for a fair comparison.

| Methods | R@1 | R@5 | R@10 | MdR | MnR |
|---|---|---|---|---|---|
| Baseline w/o alignment | 46.9 | 72.8 | 82.2 | 2.0 | 14.3 |
| Pretrained Diffusion w/ $\mathcal{L}_1$ | 30.9 | 59.5 | 68.4 | 4.0 | 40.6 |
| Pretrained Diffusion w/ $\mathcal{L}_2$ | **35.4** | **66.7** | **78.3** | **3.0** | **14.4** |

## A.3    More Details on the Proposed Method

**Discussion on $L_1$ Loss**. We perform experiments on different diffusion loss of $\mathcal{L}_1$ and $\mathcal{L}_2$ losses for the diffsuion-based alignment pretrainig. As shown in Table 7, the $\mathcal{L}_2$ provides a better performance.

Table 8: Discussion on the timestamps ($T'$) on MSRVTT.

| $T'$ | R@1 ↑ | R@5 ↑ | R@10 ↑ | MdR ↓ | MnR ↓ |
|---|---|---|---|---|---|
| 0 | 46.9 | 72.8 | 82.2 | 2.0 | 14.3 |
| 1 | 49.2 | 75.0 | 85.1 | 2.0 | 11.3 |
| 5 | 49.6 | 75.1 | 84.9 | 2.0 | **11.2** |
| 10 | **51.9** | 75.7 | 84.6 | **1.0** | 11.6 |
| 15 | 51.1 | **76.1** | **85.8** | 1.0 | 11.8 |
| 20 | 47.0 | 73.9 | 83.7 | 2.0 | 12.2 |
| 30 | 32.5 | 63.5 | 76.0 | 3.0 | 16.2 |
| 40 | 0.2 | 0.5 | 1.0 | 493.5 | 497.9 |

Table 9: Modality gap (measured by $L_1$-norm), similarity, and performance change discussion.

| Methods | Averaged Modality Gap (↓) | Averaged Similarity (↑) | R@1 (↑) |
|---|---|---|---|
| DITS fix CLIP | 23.76 | 0.122 | 39.2 |
| DITS (Ours) | 18.13 | 0.168 | 51.9 |

$\mathcal{L}_2$ loss penalizes larger errors more heavily than smaller ones because the error term is squared. This can be beneficial in tasks where it is crucial to reduce large deviations, as it forces the model to pay more attention to outliers or significant errors. By comparison, $\mathcal{L}_2$ loss treats all errors equally, which might not be ideal if large errors need to be minimized more aggressively. In the context of our problem, $\mathcal{L}_2$ ensures that these outliers are brought into alignment, reducing their impact on the overall model performance, which might be a reason for this observation.

**More Discussions on Timestamp**. In Table 8, we provide a more thorough comparison of different timestamps. As we gradually enlarge the total number timestamp $T'$ from 1 to 40, all metrics will first get better, and then decrease. This is because DITS performs a progressive alignment. It gradually narrows down the retrieval scope when we initially increase $T'$, enabling generally improved performances (R@1, R@5, MdR). However, the model loses the generalization ability if we continually inlarge $T'$, causing a performance degradation (*e.g.*, from 15 to 40). We also notice there are some inconsistent tendencies, such as R@10 decreases from $T' = 1$ to $T' = 15$, contrary to the tendency of R@1 and R@5. This is because different recall precision (*e.g.*, 1,5,10) can be regarded as different retrieval tasks with different focus. When the model continually narrows down the retrieval scope for the given query, it can easily affect R@10 that requires a larger scope. To balance between different Recall precision, we choose $T' = 10$ for MSRVTT. This experiment shows the potential of the progressive alignment in adjusting the retrieval scope. Note that devising an automated method for timestamp selection may improve performance. However, our experimental results suggest that setting the timestamp to a small range, e.g., $1 \sim 15$ yields reasonable performances and within this small range, a general and consistent setting ($T' = 10$) enables leading performances across diverse datasets, e,g, MSRVTT, LSMDC, Charades, and VATEX. Thus, we adopted the empirical timestamp setting to retain computational efficiency and method extensibility.

**Effectiveness on Learning the Modality Gap** In Table 9, we study the effect of DITS on CLIP embedding space, to uncover how DITS bridge the modality gap. We statistically compare DITS with a "DITS fix CLIP" baseline, in terms of the averaged modality gap, averaged similarity, and the performance. Specifically, we compute the $L_1$-norm of the modality gap (cosine similarity) for each relevant pair and compute the averaged value. Overall, DITS can bridge the modality gap by effectively aligning the CLIP embedding space, yielding better performance. In Fig. 5, we further provide the distribution of the similarity change. Note that Fig. 4 computes the $L_2$-norm of the modality gap distribution, corresponding to the Euclidean distance. The values (such as mean around 1.28) are scaled by $L_2$-norm and are non-bounded. We also provide the mean absolute distance version (using $L_1$-norm) of the same data in Fig. 6 (left). As shown, $L_1$-norm enables a different scale and is also non-bounded. The difference between "Joint Train" and "Fix CLIP" is more remarkable under the $L_1$-norm scale. We also provide the cosine similarities version of the same data in Fig. 6 (right). As shown, cosine similarities are bounded (i.e., ). The similarity values are generally enlarged with joint train, being consistent with Fig. 4 and Fig. 6 (left).

## A.4 Computational Cost and Efficiency

In this section, we discuss the computational efficiency for both training (Table 10) and inference (Table 11). The proposed DITS does not need much additional resources for training and reports comparable inference efficiency. We specifically compare with both non-diffusion methods (e.g., X-

Table 10: Training resource usage comparison with on MSRVTT dataset.

| Methods | GPU Memory (MB) | GPU Request | Training Time (h) |
|---|---|---|---|
| X-Pool [Gorti et al., 2022] | 18986 | 1× RTX3090 | 14.67 |
| DiffusionRet (Stage 1) | 27255 | – | 105.50 |
| DiffusionRet (Stage 2) | 8654 | – | 106.85 |
| DiffusionRet (Total) [Jin et al., 2023] | 27255 | 2× RTX3090 | 212.35 |
| T-MASS [Wang et al., 2024a] | 20390 | 1× RTX3090 | 14.74 |
| DITS (Ours) | 20950 | 1× RTX3090 | 14.90 |

Table 11: Inference time efficiency and GPU usage comparison on MSRVTT dataset.

| Methods | GPU Memory (MB) | Inference Time (s) | R@1 |
|---|---|---|---|
| X-Pool [Gorti et al., 2022] | 5452 | 65.35 | 46.9 |
| TS2-Net [Liu et al., 2022b] | 2835 | 119.50 | 47.0 |
| DiffusionRet [Jin et al., 2023] | 3375 | 64.23 | 49.0 |
| T-MASS [Wang et al., 2024a] | 5452 | 76.41 | 50.2 |
| DITS (Ours) | 5464 | 70.17 | 51.9 |

Pool, TS2-Net, T-MASS) and diffusion-based method (e.g., DiffusionRet). All training and inference costs are measured with the same computational platform (2 NVIDIA RTX3090 GPU-24GB, Intel i9-10900X CPU).

For the training, since the proposed truncated sampler DITS starts from an pre-aligned text embedding and empirically needs small numbers of iterations, it requires much smaller training time compared with conventional diffusion model-based method. Besides, DITS enjoys a comparable GPU memory usage, being easily deployed on a single GPU.

For inference, DITS requires comparable GPU memory usage and inference efficiency with previous methods. Since there is no need to refer to the video embedding at each iteration of the alignment, we can compute and cache all the aligned embeddings beforehand. Thus we can skip the iterative sampling when performing retrieval. We find this method makes DITS faster than non-diffusion methods such as TS2-Net and T-MASS.

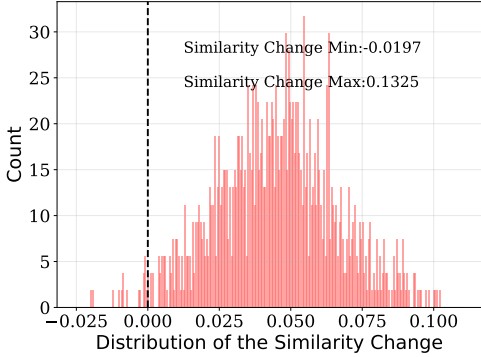

Figure 5: Similarly value subtraction between DITS and "DITS fix CLIP" (relevant pairs). DITS enables higher similarity values, yielding positive-valued histogram shown above. By comparison, DITS can effectively bridge the modality gap by aligning the CLIP embedding space.

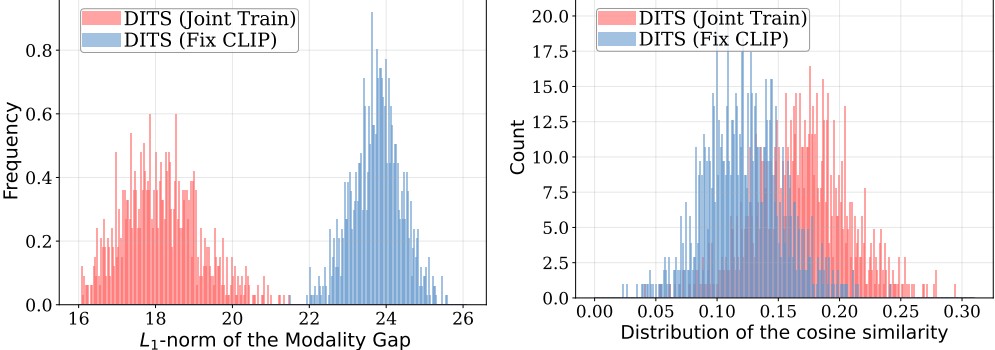

Figure 6: DITS reduce the modality gap for the **relevant** pairs. *Left*: The modality gap comparison upon $L_1$-norm (Mean absolute distance, unbounded). *Right*: The modality gap comparison measured by the cosine similarity (bounded, i.e., $[0, 1]$).

