# OpenReview forum: "Diffusion-Inspired Truncated Sampler for Text-Video Retrieval"
_NeurIPS.cc/2024/Conference — NeurIPS 2024 poster_

### Official Review · Reviewer_vaB9 · 2024-07-06

**Soundness:** 3
**Presentation:** 2
**Contribution:** 3
**Rating:** 6
**Confidence:** 4

**Summary:**

The paper introduces a new method, Diffusion-Inspired Truncated Sampler (DITS), designed for text-video retrieval tasks. It addresses the primary challenge of bridging the modality gap between text and video data, a problem that existing retrieval methods often fail to solve effectively. The authors propose DITS to harness the strengths of diffusion models in reducing this gap and enhancing the alignment between text and video embeddings within a joint embedding space.

**Strengths:**

* The authors investigate the use of Diffusion models for addressing the text-video modality gap and identify limitations in applying standard Diffusion models to retrieval tasks. Specifically, they highlight that the L2 loss is not well-suited for the ranking problems inherent in retrieval, and that there is a dependency on varied initial points from an isotropic Gaussian distribution.
* Extensive experiments on five benchmark datasets demonstrate DITS's state-of-the-art performance. The method shows flexibility in adjusting the retrieval scope over time and improves the structure of the CLIP embedding space.
* The authors have committed to releasing the code, which will facilitate further research and application of the proposed method.

**Weaknesses:**

* Generalization ability. The authors highlight the importance of selecting an optimal number of timestamps for the truncated diffusion process. If the number of timestamps is too large, the model might lose its generalization ability.
* Computational efficiency. Computational efficiency is a common concern with Diffusion models, which typically require substantial computational resources for training and inference.
* There are some confusions in the writing. In Section 3.2, it is mentioned that the condition for diffusion is the textual feature. However, in Section 3.3, the condition is not specified. Additionally, in Appendix A.3, the authors state that an empty condition works best.

**Questions:**

* The authors would do well to make a comparison of inference times.
* Since the timestamps for the truncated diffusion process have a huge impact on performance, could an automated method be devised to determine time steps?
* The performance of unconditional diffusion surpasses that of text or video diffusion, which is counter-intuitive. Could the authors provide further analysis to explain this phenomenon?

**Limitations:**

The limitations of the method have been discussed in detail.

---

> ### Author Rebuttal · Authors · 2024-08-05
>
> We much appreciate that Reviewer `vaB9` provides valuable comments and finds the proposed method shows state-of-the-art  performance. We are committed to release the training and inference code, as well as the pretrained models.
>
> **`R5.1`**: Generalization ability. The authors highlight the importance of selecting an optimal number of timestamps for the truncated diffusion process. If the number of timestamps is too large, the model might lose its generalization ability.
>
> **`A5.1`**: Thanks for the valuable comment. Since CLIP embedding space offers a good proximity between text and video (L57\~L58) as shown in prior works [R1, R2], the timestamp is not supposed to be too large. We also empirically find that DITS
>  achieves good performance with a small timestamp, e.g., $T’=10$ on MSRVTT, LSMDC, Charades, and VATEX (L270). Both the previous study [R1, R2] and empirical results suggest that DITS retains a good generalization ability using a relatively small and consistent timestamp setting, being free from the elaborate tuning.
>
> [R1] Mind the Gap: Understanding the Modality Gap in Multi-modal Contrastive Representation Learning. In NeurIPS 2022.
>
> [R2] Shifted diffusion for text-to-image generation. In CVPR 2023.
>
> **`R5.2`**: Computational efficiency. Computational efficiency is a common concern with Diffusion models, which typically require substantial computational resources for training and inference.
>
> **`A5.2`**: Thanks for the valuable suggestion. We provide a thorough discussion on the computational efficiency including runtime, resource usage for both training (`Table T1`) and inference (`Table T2`) in the `Global Rebuttal`. Overall, DITS requires comparable GPU resources and time for training (e.g., DITS: 14.90h v.s. DiffusionRet: 212.35h). DITS also reports comparable inference efficiency, such as compared to previous methods of TS2-Net (DITS: 70.17s v.s. TS2-Net: 119.50s) and T-MASS (DITS: 70.17s v.s. T-MASS: 76.41s).
>
> **`R5.3`**: There are some confusions in the writing. In Section 3.2, it is mentioned that the condition for diffusion is the textual feature. However, in Section 3.3, the condition is not specified. Additionally, in Appendix A.3, the authors state that an empty condition works best.
>
> **`A5.3`**: Thanks for the detailed proofreading. For the Diffusion model baseline (Section 3.2), the condition is the textual feature. For the proposed DITS (Section 3.3, Appendix A.3), no condition is needed (empty condition). To clarify, the conventional Diffusion model (Section 3.2, proposed baseline) has to adopt the text condition to guide the learning. Differently, based on the experiment in Appendix A.3, we find an empty condition for DITS works best, thus does not specify condition for DITS in Section 3.3. Notably,  DITS and conventional Diffusion models adopt the same text embedding in different ways, DITS  takes the text embedding as the starting point, and the diffusion model takes text embedding as the condition. We will make it more clear by pointing out in Section 3.3 that the condition is not used in DITS.
>
> **`R5.4`**: Since the timestamps for the truncated diffusion process have a huge impact on performance, could an automated method be devised to determine time steps?
>
> **`A5.4`**: Thanks for the valuable comment. Devising an automated method for timestamp selection can be an attractive direction and may improve performance. However, our experimental results suggest that setting the timestamp to a small range, e.g., $1\sim15$ yields reasonable performances (Table 7) and within this small range, a general and consistent setting ($T’=10$) enables leading performances across diverse datasets, e,g, MSRVTT, LSMDC, Charades, and VATEX (L270). Thus, we adopted the empirical timestamp setting to retain computational efficiency and method expansibility.
>
> We could provide a potential solution with a bi-level optimization framework. The upper-level can optimize the timestamp hyperparameter and the lower-level can take the timestamp to guide the iterative alignment. One can potentially perform an alternating training strategy with contrastive loss. However, a series of challenges can persist: timestamps may affect the iterative diffusion process non-linearly, introducing the local minima or saddle points. The search space for timestamps can be high-dimensional, considering the broad context on which the timestamp operates. Evaluating different timestamps can be computationally intensive.
>
> Overall, DITS provides a new viewpoint and foundation for future research and we will continue exploring per reviewer’s inspiration. We will add this in the manuscript.
>
> **`R5.5`**: The performance of unconditional diffusion surpasses that of text or video diffusion, which is counter-intuitive. Could the authors provide further analysis to explain this phenomenon?
>
> **`A5.5`**:  Thanks for the valuable suggestion. The diffusion process for this task emphasizes ``accurate alignment’’, unlike the general Diffusion models highlighting diversity in creative works. The previous intuition of the diffusion condition may not be applicable in this work. We provide further analysis to elaborate this phenomenon:
> * Due to the one-to-many mapping of video-to-text, one video can map to multiple gap vectors. Taking the video as a condition can guide the model learn the undesired modality gap (L599\~L601), empirically dropping the performance.
> * DITS start from the text, ensuring the alignment starts from a semantically meaningful initial point. The text condition acts as the constraint or guideline for the learning. Simultaneously applying the text embedding as the starting point and the condition can cause conflicting instructions, empirically decreasing the performance.
> * Joint use of text and video conditions can inherent both limitations, empirically leading to an unsatisfactory performance.
>
> We will incorporate the above analysis in the final version.

---

> > ### Comment · Reviewer_vaB9 · 2024-08-10
> >
> > Thank you for your detailed reply. My concerns have been addressed. I have also read the comments from the other reviewers and the authors' responses. I think the paper as a whole is interesting, though it would benefit from more detailed descriptions and discussions. Therefore, I will keep my rating.

---

> > > ### Author Response · Authors · 2024-08-10
> > > **Response To Reviewer vaB9**
> > >
> > > We appreciate the reviewer's recognition of our response and support for our work!

---

### Official Review · Reviewer_UKTF · 2024-07-10

**Soundness:** 4
**Presentation:** 4
**Contribution:** 4
**Rating:** 7
**Confidence:** 5

**Summary:**

This paper smartly leverage the diffusion model to solve a famous modality gap problem in the a CLIP retrieval problem. The proposed method called DITS set the text embedding as the initial point in the 1D diffusion model and try to generate the video embedding. Finally, by using the contrastive loss to align the generated video embedding and ground truth video embedding to finish the training of the diffusion model. The work show convincible result comparing to existing SOTA methods and the ablation study in Table 4 and 5 show the effectiveness of the method comparing to raw CLIP and existing work DiffusionRet.

**Strengths:**

The idea to leverage the diffusion model to solve modality gap is highly novel.
It is a nice viewpoint to the famous modality gap problem.
After this paper, the problem will highly to be reexamined in the community.
1. The gap should be some intrinsic problem in the multi-modality model training.
2. With this paper, the gap is kind of filled and most of old retrieval methods and work will be re-actived in this research area.

**Weaknesses:**

The figure is too small to read.(miner issue)
Figure 4, it looks like the gap is still large as the mean absolute distance is ~1.28. why it is not some small value likes 0.5?
Is there any investigation on cases, which video retrieval cases got improved and which video cases become worse?

**Questions:**

check the weakness

---

> ### Author Rebuttal · Authors · 2024-08-05
>
> We much appreciate that Reviewer `UKTF` provides valuable comments and finds the idea is novel, showing effectiveness in  filling the gap.
>
> **`R4.1`**: The figure is too small to read.(miner issue) Figure 4, it looks like the gap is still large as the mean absolute distance is ~1.28. why it is not some small value likes 0.5?
>
> **`A4.1`**: Thanks for the valuable comment. We will revise the layout of the figure to make it more clear. In the manuscript, Fig. 4 computes the $L\_2$-norm of the modality gap distribution, corresponding to the Euclidean distance. The values (such as mean around 1.28) are scaled by $L\_2$ norm and are non-bounded.
>
> We also provide the mean absolute distance version (using $L\_1$-norm) of the same data in `Fig.F2` (left) in the attached rebuttal PDF.  As shown, $L\_1$-norm enables a different scale and is also non-bounded. The difference between “Joint Train” and “Fix CLIP” is more remarkable under the $L\_1$-norm scale.
>
> We also provide the cosine similarities version of the same data in `Fig. F2` (right) in the attached rebuttal PDF. As shown, cosine similarities are bounded (i.e., $[0,1]$). The similarity values are generally enlarged with joint train, being consistent with Fig. 4 and `Fig. F2` (left). We will put above evidence and analysis into the manuscript and provide more illustrations to the values in Fig. 4.
>
> **`R4.2`**: Is there any investigation on cases, which video retrieval cases got improved and which video cases became worse?
>
> **`A4.2`**: Thanks for the valuable comment. We provide investigations and potential intuitions on both retrieval cases when comparing DITS with the raw CLIP baseline in Fig. 4. (1) Since DITS performs the iterative alignment, it can better avoid drastic changes and provide more chances to adjust the misalignment step. We find when the text and video data is more challenging (such as vague texts or blurry videos), DITS can identify the relevant pairs and get the performance improved. (2) Conversely, on the simpler cases where text is rich and informative, or the video is temporally consistent, we did not see a large difference between DITS and the raw CLIP baseline. However, we observe when the caption data is incorrectly annotated, DITS can have a chance to miss the relevant video. We will add the above discussions into the manuscript.

---

### Official Review · Reviewer_QeV9 · 2024-07-12

**Soundness:** 2
**Presentation:** 2
**Contribution:** 2
**Rating:** 4
**Confidence:** 4

**Summary:**

The authors introduce Diffusion-Inspired Truncated Sampler  (DITS) that jointly performs progressive alignment and modality gap modeling in the joint embedding space.  Experiments on five benchmark datasets suggest the state19 of-the-art performance of DITS.

**Strengths:**

1. The motivation is clearly described and easy to understand.
2. This work studies the Diffusion model to bridge the modality gap of text-video retrieval, identifying two key limitations of the vanilla Diffusion model.
3. Extensive experiments on five datasets (MSRVTT, LSMDC, DiDeMo, VATEX, and Cha76 rades) suggest that DITS achieves state-of-the-art performance.

**Weaknesses:**

1. In Table 1, under the CLIP-ViT-B/32 feature extractor, the author's method has limited performance improvement compared to the comparison method.
2. In Formula 2, these two terms are added, but why are these two the same? They should be different terms added.
3. What does it mean to multiply ϵ, t, c in the norm of Formula 6? This is not standard.
4. The writing of the methods section needs further standardization and improvement.

**Questions:**

1. In Table 4, do Diffusion and Fine tune parts not need L2 loss? Why does Pretrain have L2 loss, while Fine tune does not need L2 loss and the result is high?
2. In Table 4, does the last row DITS not need L2 loss?

**Limitations:**

See the weaknesses section

---

> ### Author Rebuttal · Authors · 2024-08-05
>
> We much appreciate that Reviewer `QeV9` finds the motivation is clear and easy to follow, and the proposed method achieves the state-of-the-art performance.
>
> **`R3.1`**: In Table 1, under the CLIP-ViT-B/32 feature extractor, the author's method has limited performance improvement compared to the comparison method.
>
> **`A3.1`**: (1) For the performance comparison, other methods can use larger settings than DITS (L283\~286), e.g., larger batch sizes such as 64 or 128 to help improve their performance. Due to limited computational resources, DITS adopts batch size of 32
>  (L274) and achieves a remarkable boost, especially compared with the previous diffusion-based method, DiffusionRet (+2.9\% at R@1 on MSRVTT, +4.4\% at R@1 on DiDeMo). (2) DITS enables better scalability. The boost over T-MASS is enhanced (MSRVTT: 2.3\% at R@1, LSMDC: 0.7\% at R@1) when changing CLIP-ViT-B/32 to CLIP-ViT-B/16.
>
> **`R3.2`**: In Formula 2, these two terms are added, but why are these two the same? They should be different terms added.
>
> **`A3.2`**: The two terms in the Eq.2 are different. We follow the standard notations of symmetric-formed cross entropy loss (specifically, InfoNCE loss [R1]) that is used in text-video retrieval domain (including both text-to-video and video-to-text) [R2, R3, R4, etc.]. To clarify, we directly copied it from the manuscript and highlight the difference between two terms below:
>
> $\mathcal{L}\_{\texttt{sce}} = - \frac{1}{B}\sum\limits^{B}\_{i=1}\log\frac{e^{s(\mathbf{t}^{(i)}, \mathbf{v}^{(i)})\cdot \tau}}{\sum\nolimits\_{\textcolor{red}{j}}e^{s(\mathbf{t}^{(i)}, \mathbf{\textcolor{red}{v}}^{\textcolor{red}{(j)}})\cdot \tau}} + \log\frac{e^{s(\mathbf{t}^{(i)}, \mathbf{v}^{(i)})\cdot \tau}}{\sum\nolimits\_{\textcolor{red}{j}}e^{s(\mathbf{\textcolor{red}{t}}^{\textcolor{red}{(j)}}, \mathbf{v}^{(i)})\cdot \tau}}$
>
> * The first term makes a summation over $v^{(j)}$ in the denominator, representing the sum of similarities between the i-th query text $\mathbf{t}^{(i)}$ and all key videos $\mathbf{v}^{(j)}$
> * The second term makes a summation over $t^{(j)}$ in the denominator, representing the sum of similarities between the i-th query video $\mathbf{v}^{(i)}$ and all texts $\mathbf{t}^{(j)}$.
>
>  We will put more descriptions into the manuscript and are willing to illustrate any point that remains unclear to the reviewer.
>
> [R1] Representation Learning with Contrastive Predictive Coding.
>
> [R2]  X-pool: Cross-modal language-video attention for text-video retrieval. In CVPR 2022.
>
> [R3] Unified Coarse-to-Fine Alignment for Video-Text Retrieval. In ICCV 2023.
>
> [R4] Text is mass: Modeling as stochastic embedding for text-video retrieval. In CVPR 2024.
>
> **`R3.3`**: What does it mean to multiply ϵ, t, c in the norm of Formula 6? This is not standard.
>
> **`A3.3`**: We kindly remind the reviewer that $\textcolor{red}{\epsilon}$, $\textcolor{red}{t}$, and $\textcolor{red}{c}$ are not multiplied in Eq.6. $\textcolor{red}{\epsilon}$, $\textcolor{red}{t}$, and $\textcolor{red}{c}$ are three inputs of the denoising network $\textcolor{blue}{\epsilon\_\gamma(\cdot)}$ (L165, L172). As we directly copied from the manuscript and highlight below,
>
> $\mathcal{L}\_\gamma = \mathbb{E}\_{\mathbf{\delta}\_0, t, \epsilon} [||\epsilon - \textcolor{blue}{\epsilon\_\gamma(}\sqrt{\bar{\alpha}}\mathbf{\delta}\_0 + \sqrt{1-\bar{\alpha}\_t}\textcolor{red}{\epsilon}, \textcolor{red}{t}, \textbf{\textcolor{red}{c}}\textcolor{blue}{)}||^2]$
>
> To clarify, (1) using $\epsilon$ to denote the noise (or the corresponding denoising network), $t$ to denote the timestamp, and $c$ to denote the condition are standard representation forms in Diffusion model literatures [R5, R6, R7]. We will add more descriptions toward Eq.6 in the manuscript. We are willing to provide more illustrations to help address further concerns of the reviewer.
>
> [R5] Denoising diffusion probabilistic models. In NeurIPS 2020.
>
> [R6] Adding conditional control to text-to-image diffusion models." In ICCV 2023.
>
> [R7] Noise2Music: Text-conditioned Music Generation with Diffusion Models. Google Research.
>
> **`R3.4`**: The writing of the methods section needs further standardization and improvement.
>
> **`A3.4`**: We appreciate the reviewer’s commitment in helping improve the manuscript. We will consider all comments of the reviewer and carefully revise the method section accordingly.
>
> **`R3.5`**: In Table 4, do Diffusion and Fine tune parts not need L2 loss? Why does Pretrain have L2 loss, while Fine tune does not need L2 loss and the result is high? In Table 4, does the last row DITS not need L2 loss?
>
> **`A3.5`**: Thanks for the valuable comment. “Diffusion pretrain” adopts conventional $L\_2$ loss. “Diffusion Fine tune” adopts the contrastive loss $L\_{sce}$ (L315) instead of $L\_2$ loss.  We make it more clear  about the loss for different baselines in Table 4 in the manuscript, as copied in `Table T3` in the attached rebuttal PDF. We provide comments below to help the reviewer better locate the key points concerning the ablation study.
>
> * **Diffusion Pretrain**: We firstly study the effectiveness of vanilla Diffusion model with $L\_2$ loss in text-video retrieval, identifying the limitation of $L\_2$ loss (abstract L8\~L10, introduction L48\~L51, method L210\~L218, and experiment L312\~L314).
> * **Diffusion Fine tune**: Based on the above observation, we fine tune the Diffusion model with contrastive loss ($L\_{sce}$ in  Eq.2), obtaining a performance boost (L315). Contrastive loss jointly considers relevant and irrelevant pairs, calibrating the embedding that is misaligned by the $L\_2$ loss.
> * **DITS**: Based on the above observation, DITS adopts contrastive loss and does not need $L\_2$ loss (abstract: L16, introduction L62\~L64, method L238\~L240).
>
> We are willing to provide more illustrations in solving the reviewer’s further concerns.

---

### Official Review · Reviewer_fBRP · 2024-07-12

**Soundness:** 4
**Presentation:** 4
**Contribution:** 4
**Rating:** 7
**Confidence:** 4

**Summary:**

The paper tackles the task of text-video retrieval. It aims to address the modality gap between text and video that usually stems in state-of-the-art models. In order to do this, it leverages Diffusion models. So, it introduces DITS that jointly performs progressive alignment and modality gap modeling in the joint embedding space in order to improve the performance. Finally, the authors test the performance of their method on five benchmarks.

**Strengths:**

The paper tackles an important task and achieves good results. I find the idea interesting and the paper is fairly well written.

**Weaknesses:**

While I think that the paper is fairly well written, some parts can be a bit confusing at the first read, though they become clear if you read twice. For example, abstract lines 6-8 first states that Diffusion is used to mitigate the problem, but then there is immediately the claim that there are "flaws" in diffusion. So, I think a rephrasing and introducing a bit later that changes are needed to diffusion in order to make that work would be better.

line 145 "Unfortuntely, we find descent retrieval performances" -> "Unfortunately, there is a decrease in retrieval performance"

The main concern that I have is related to understanding the limitations. I think there should have been a section that discusses the need for additional computational resources, if applicable and how does the running time compare agains non diffusion methods.

**Questions:**

Is there any additional cost in terms of running time for the proposed method as opposed to other sota methods?

**Limitations:**

The limitations are discussed in the appendix.

---

> ### Author Rebuttal · Authors · 2024-08-05
>
> We much appreciate that Reviewer `fBRP` provides valuable comments and finds the idea is novel and interesting, with good results.
>
> **`R2.1`**: While I think that the paper is fairly well written, some parts can be a bit confusing at the first read, though they become clear if you read twice. For example, abstract lines 6-8 first states that Diffusion is used to mitigate the problem, but then there is immediately the claim that there are "flaws" in diffusion. So, I think a rephrasing and introducing a bit later that changes are needed to diffusion in order to make that work would be better.
>
> **`A2.1`**: Thanks for the reviewer’s commitment in helping improve the manuscript. We rephrase the abstract lines according to the suggestion as follows: ``In this work, we leverage the potential of Diffusion models to address the text-video modality gap by progressively aligning text and video embeddings in a unified space. However, we identify two key limitations of existing Diffusion models in retrieval tasks.’’. We will accordingly revise the introduction to ensure better clarity and coherence. We are committed to making further modifications if any parts of the manuscript remain unclear.
>
> **`R2.2`**: line 145 "Unfortunately, we find descent retrieval performances" -> "Unfortunately, there is a decrease in retrieval performance"
>
> **`A2.2`**: Thanks for the detailed proofreading. We will change this sentence in the manuscript as suggested by the reviewer.
>
> **`R2.3`**: The main concern that I have is related to understanding the limitations. I think there should have been a section that discusses the need for additional computational resources, if applicable and how does the running time compare against non diffusion methods.
>
> **`A2.3`**: Thanks for the valuable suggestion. We provide a thorough discussion on the computational efficiency including runtime, resource usage for both training (`Table T1`) and inference (`Table T2`) in the `Global Rebuttal`. Overall, DITS does not need much additional resources for training and reports comparable efficiency especially with non-diffusion methods.
>
> (1) As DITS starts from the text embedding pre-aligned by CLIP and empirically needs small numbers of iterations,  it does not need much additional cost for the training. Specifically, DITS requires comparable GPU usage and time usage compared with non-diffusion methods, including X-Pool, TS2-Net, and T-MASS. (2) For inference, DITS requires comparable GPU memory usage and inference runtime -- as there is no need to refer to the video embedding at each iteration of the alignment, we can compute and cache all the aligned embeddings beforehand and skip the iterative sampling when performing retrieval. We find this implementation strategy makes DITS faster than non-diffusion methods such as TS2-Net (DITS: 70.17s *v.s.* TS2-Net: 119.50s) and T-MASS (DITS: 70.17s *v.s.* T-MASS: 76.41s). We will add the above discussions into the final version.

---

> > ### Comment · Reviewer_fBRP · 2024-08-10
> > **Rebuttal answer**
> >
> > Thank you for providing additional details! I confirm that I read the rebuttal and I don't currently have other questions

---

> > > ### Author Response · Authors · 2024-08-10
> > > **Response To Reviewer fBRP**
> > >
> > > We appreciate the reviewer's valuable comments. We thank the reviewer's recognition of our rebuttal!

---

### Official Review · Reviewer_CwRd · 2024-07-14

**Soundness:** 3
**Presentation:** 3
**Contribution:** 2
**Rating:** 4
**Confidence:** 4

**Summary:**

The paper introduces a novel method to address the challenge of bridging the modality gap between text and video data in retrieval tasks. The authors propose the Diffusion-Inspired Truncated Sampler (DITS), leveraging diffusion models to model the text-video modality gap.
DITS performs progressive alignment and modality gap modeling, starting from text embeddings and using a truncated diffusion process to generate aligned video embeddings.

**Strengths:**

1. The paper introduces a novel method, DITS, which leverages diffusion models to address the modality gap in text-video retrieval. The authors have identified and addressed the limitations of existing diffusion models when applied to ranking tasks, which is a significant contribution to the field.

2. The authors have provided a theoretical foundation for their method, including a discussion on the limitations of L2 loss and the benefits of truncated diffusion processes.

**Weaknesses:**

1. The author claims that the vanilla Diffusion model's L2 loss does not fit the ranking problem in text-video retrieval. However, in the method part (Line 183), the author still uses the conventional L2 loss of diffusion model. I want to know the detail of that how the author addresses the problem of L2 loss.

2. As far as I know, it is unrealistic to directly use the diffusion model to learn the modality gap or the joint distribution probability of cross-modal alignment. For example, after I ran through the code of [1], I found that its main contribution is not in the diffusion model. Like [2], using the diffusion model to solve the impact of time distribution on moment retrieval is a reasonable and effective motivation. Therefore, the author needs to provide more quantitative experiments to prove that the proposed diffusion model can learn the accurate modality gap, such as similarity change or other metrics.

3. The paper could provide more details on the computational efficiency of the proposed method, including runtime and resource usage, which are important considerations for practical deployment. According to my experience of using diffusion models to solve retrieval tasks, the diffusion module will seriously slow down the retrieval speed, and has limited improvement in optimizing alignment quality and improving retrieval accuracy. The experimental results of this paper also show that the proposed method has limited improvement over the SOTA method. The author should provide experimental data on time efficiency. I think this improvement is not enough compared to the obvious lack of time efficiency.

4. The author assumes that the modality gap is Gaussian distributed. Is there any corresponding proof or basis for this assumption?

[1] Diffusionret: Generative text-video retrieval with diffusion model. In ICCV, 2023.
[2] MomentDiff: Generative Video Moment Retrieval from Random to Real

**Questions:**

1. The author claims that the vanilla Diffusion model's L2 loss does not fit the ranking problem in text-video retrieval. However, in the method part (Line 183), the author still uses the conventional L2 loss of diffusion model. I want to know the detail of that how the author addresses the problem of L2 loss.

2. As far as I know, it is unrealistic to directly use the diffusion model to learn the modality gap or the joint distribution probability of cross-modal alignment. For example, after I ran through the code of [1], I found that its main contribution is not in the diffusion model. Like [2], using the diffusion model to solve the impact of time distribution on moment retrieval is a reasonable and effective motivation. Therefore, the author needs to provide more quantitative experiments to prove that the proposed diffusion model can learn the accurate modality gap, such as similarity change or other metrics.

3. The paper could provide more details on the computational efficiency of the proposed method, including runtime and resource usage, which are important considerations for practical deployment. According to my experience of using diffusion models to solve retrieval tasks, the diffusion module will seriously slow down the retrieval speed, and has limited improvement in optimizing alignment quality and improving retrieval accuracy. The experimental results of this paper also show that the proposed method has limited improvement over the SOTA method. The author should provide experimental data on time efficiency. I think this improvement is not enough compared to the obvious lack of time efficiency.

4. The author assumes that the modality gap is Gaussian distributed. Is there any corresponding proof or basis for this assumption?

[1] Diffusionret: Generative text-video retrieval with diffusion model. In ICCV, 2023.
[2] MomentDiff: Generative Video Moment Retrieval from Random to Real

**Limitations:**

See Weaknesses part.

---

> ### Author Rebuttal · Authors · 2024-08-05
>
> We much appreciate that Reviewer `CwRd` provides valuable comments and finds the proposed method novel. Since the weakness and questions are the same, we answer the weakness below. Due to the limited rebuttal space, we will not be able to copy all questions but summarize the weakness statements below.
>
> **`R1.1`**: The author claims that the vanilla Diffusion model's L2 loss does not fit the ranking problem in text-video retrieval. However, in the method part (Line 183), the author still uses the conventional L2 loss of diffusion model. I want to know the detail of that how the author addresses the problem of L2 loss.
>
> **`A1.1`**: We use the contrastive loss ($\mathcal{L}\_\text{sce}$ in Eq.2)  but not $\mathcal{L}\_2$ loss to train DITS (see L239). The method part (L183, Section 3.2) is a baseline method, not our full model DITS. The statement of “conventional $\mathcal{L}\_2$ loss does not fit the ranking problem in text-video retrieval” is based on this baseline that uses the $\mathcal{L}\_2$ loss of the diffusion model. To elaborate, $\mathcal{L}\_2$ loss only minimizes the modality gap between the relevant pairs (as shown in Fig.3 in the manuscript), failing to handle irrelevant pairs. To address the $\mathcal{L}\_2$ loss issue, we adopt the contrastive loss $\mathcal{L}\_{sce}$ to incorporate with a proposed truncated sampler. The resulting method DITS models the gap in an iterative manner and achieves promising retrieval performance.
>
> **`R1.2`**: Provide more quantitative experiments to prove that the proposed diffusion model can learn the accurate modality gap, such as similarity change or other metrics.
>
> | Methods | Averaged modality gap ($\downarrow$) | Averaged Similarity ($\uparrow$) | R@1 ($\uparrow$) |
> |----------|----------|----------|----------|
> | DITS fix CLIP | 23.76 | 0.122 | 39.2 |
> | DITS (Ours) | **18.13** | **0.168** | **51.9** |
>
> **Table T4. Modality gap (measured by $L\_1$-norm), similarity, and performance change discussion on DITS.**
>
> **`A1.2`**: Previous methods [DiffusionRet (ICCV 2023), MomentDiff (NeurIPS 2023)]  learn a mapping from random noise to the signal (e.g., real moments) using the diffusion  $\mathcal{L}\_1$  loss. Differently, DITS adopts the contrastive loss $\mathcal{L}\_\text{sce}$ instead of $\mathcal{L}\_1$ (or $\mathcal{L}\_2$) loss. DITS starts from the text embedding with the proposed truncated sampler instead of from the random noise. These differences enable DITS to iteratively learn the modality gap and distinguish from previous diffusion-based designs.  We will include and discuss MomentDiff [2] in the related work.
>
> In `Table T4` above,  we study the effect of DITS on CLIP embedding space, to uncover how DITS bridge the modality gap. We statistically compare DITS with a ``DITS fix CLIP'' baseline, in terms of the averaged modality gap, averaged similarity, and the performance. Specifically, we compute the  $L\_1$-norm of the modality gap (cosine similarity) for each relevant pair and compute the averaged value. Overall, DITS can bridge the modality gap by effectively aligning the CLIP embedding space, yielding better performance. In `Fig.F1` in the attached PDF, we further provide the distribution of the similarity change per reviewer's request.
>
> **`R1.3`**: More details on the computational efficiency including runtime and resource usage is expected.
>
> **`A1.3`**: Thanks for the valuable comment. We provide a thorough discussion on the computational efficiency comparison for both training (`Table T1`) and inference (`Table T2`) in the `Global Rebuttal`. The proposed DITS does not need much additional resources for training (e.g., DITS: 14.90h *v.s.* DiffusionRet: 212.35h) and reports comparable inference efficiency (e.g., DITS: 70.17s *v.s.* TS2-Net: 119.50s). For the performance comparison, state-of-the-art methods can use larger settings than DITS (L283\~286, e.g., larger batch sizes such as 64, 128) to improve their performance. By comparison, DITS adopts smaller batch size (i.e., 32, L274) and achieves a remarkable boost, especially compared with DiffusionRet (+2.9% at R@1 on MSRVTT, +4.4% at R@1 on DiDeMo). We will add the discussions on computational resources and efficiency into the final version.
>
> **`R1.4`**:  The author assumes that the modality gap is Gaussian distributed. Is there any corresponding proof or basis for this assumption?
>
> **`A1.4`**: Thanks for the valuable comment. We posit Gaussian distribution based on its mathematical properties and previous works in multimodal learning [R1\~R6]. Gaussian distributions offer (a) parametric convenience -- capturing the central tendency (mean) and the variability (variance) of the modality gap and (b) mathematical tractability -- linear transformation and additivity, ensuring the modality gap retains the Gaussian after the processing pipeline.  Previous works benefit from positing the multimodality gap as the Gaussian, such as (1) quantifying the multimodal uncertainty [R1, R3, R4, R6], (2) enriching the text embedding with flexible and resilient semantics [R2], and (3) enabling data augmentation via Gaussian noise injection [R5]. We also empirically observe the Gaussian-like multimodality gap in our experiment (e.g., Fig. 2, 4).  We will discuss the references in the related work.
>
> [R1] Embracing Unimodal Aleatoric Uncertainty for Robust Multimodal Fusion. In CVPR 2024.
>
> [R2] Text Is MASS: Modeling as Stochastic Embedding for Text-Video Retrieval. In CVPR 2024.
>
> [R3] Uncertainty-based Cross-Modal Retrieval with Probabilistic Representations. In CVPR 2023.
>
> [R4] MAP: Multimodal Uncertainty-Aware Vision-Language Pre-training Model. In CVPR 2023.
>
> [R5] Text-Only Training for Image Captioning using Noise-Injected CLIP. In EMNLP 2022.
>
> [R6] Probabilistic Embeddings for Cross-Modal Retrieval. In CVPR 2021.

---

### Author Rebuttal · Authors · 2024-08-05

# Global Rebuttal

We would like to express our sincere gratitude to all reviewers for the time and effort in reviewing our manuscript. We much appreciate  that reviewers find the proposed method novel and achieve good results.   We are committed to release the training and inference code, as well as the pretrained models. We will address each reviewer's concern in each rebuttal. Due to the limited space, we provide extra results below.

* We provide an overall discussion on the computational cost and efficiency below.
* We provide `Figure F1` in the attached PDF to address the concern of the similarity change for Reviewer `CwRd`.
* We provide `Table T3` in the attached PDF to make it clear for Table 4 for Reviewer `QeV9`.
* We provide `Figure F2` in the attached PDF to address the concern of Fig. 4 for Reviewer `UKTF`.

**Discussion on the Computational Cost and Efficiency**

We discuss the computational efficiency for both training (`Table T1`) and inference (`Table T2`) to solve reviewers’ concerns.  The proposed DITS does not need much additional resources for training and reports comparable inference efficiency. We specifically compare with both non-diffusion methods (e.g., X-Pool, TS2-Net, T-MASS) and diffusion-based method (e.g., DiffusionRet). All training and inference costs are measured with the same computational platform (2$\times$NVIDIA RTX3090 GPU-24GB, Intel i9-10900X CPU). We will add these discussions into the final version.

| Methods | GPU memory (MB)    | GPU Request    | Training Time (h)    |
|----------|-------------|-------------|-------------|
|X-Pool | 18986          | 1 x RTX3090           | 14.67           |
|DiffusionRet (Stage 1) | 27255           | --           | 105.50          |
|DiffusionRet (Stage 2) | 8654           | --           | 106.85           |
|DiffusionRet (Total) | 27255           | 2 x RTX3090           | 212.35           |
|T-MASS| 20390           | 1 x RTX3090           | 14.74           |
|DITS (Ours)| 20950           | 1 x RTX3090           | 14.90           |

**Table T1. Training resource usage comparison with  on MSRVTT dataset.**

| Methods | GPU memory (MB) | Inference Time (s) | R@1 |
|----------|----------|----------|----------|
| X-Pool (CVPR 2022)       | 5452        | 65.35        | 46.9        |
| TS2-Net (ECCV 2022)       | 2835        | 119.50        | 47.0        |
| DiffusionRet (ICCV 2023)       | 3375        | 64.23        | 49.0        |
| T-MASS (CVPR 2024)        | 5452        | 76.41        | 50.2        |
| DITS (Ours)        | 5464        | 70.17        | 51.9        |

**Table T2. Inference time efficiency and GPU usage comparison on MSRVTT dataset.**

For the training, since the proposed truncated sampler DITS (1) starts from an pre-aligned text embedding and (2) empirically needs small numbers of iterations (L270, $T’=10$ on four benchmark datasets),  it requires much smaller training time compared with conventional diffusion model-based method. For example, DiffusionRet requires in total over one week to finish the two stage training despite having a large batch size (i.e., 128 in DiffusionRet paper). Besides, DITS enjoys a comparable GPU memory usage, being easily deployed on a single GPU.

For inference, DITS requires comparable GPU memory usage and inference efficiency with previous methods. Since there is no need to refer to the video embedding at each iteration of the alignment, we can compute and cache all the aligned embeddings beforehand. Thus we can skip the iterative sampling when performing retrieval. We find this method makes DITS faster than non-diffusion methods such as TS2-Net and T-MASS.

---

### Decision · Program_Chairs · 2024-09-25

**Decision:**

Accept (poster)

**Comment:**

This paper received mixed reviews, with three strongly positive reviews (scores 6,7,7) and two mildly negative ones (scores 4,4).

The paper introduces a novel method to address the challenge of bridging the modality gap between text and video data in retrieval tasks. The authors propose the Diffusion-Inspired Truncated Sampler (DITS), leveraging diffusion models to model the text-video modality gap. DITS performs progressive alignment and modality gap modeling, starting from text embeddings and using a truncated diffusion process to generate aligned video embeddings.

The reviewers agreed that the problem is important, the solution innovative, the theoretical motivation solid, and that the method achieved good results. The main concerns were computational efficiency, some writing issues, and various questions of detail. The authors provided a rebuttal that address these issues. Most of the positive reviewers confirmed that the rebuttal solved all their concerns. Unfortunately, the negative reviewers chose not to participate in the post review phase. This, the fact that many of the concerns they raised seem more like requests for clarification that real negative flags, and that the explanations provided by the authors seem sensible, have led the AC to discount the negative scores.

As a note to the authors. I would say that it is not OK to accuse the reviewers of bias just because they did not review your work positively. These type of remarks do not help your cause. If the reviewer did not fully understood your paper, it is likely because the paper could better written. In fact, other reviewers have mentioned that the paper could be more clear. In the future, please do not engage in this type of behavior again.